# Asynchronous Matching with Dynamic Sampling for Multimodal Dataset Distillation

**Ding Qi**[1]   **Jian Li**[2,4]   **Shuguang Dou**[1]   **Zifan Song**[1]   **Junyao Gao**[1]   **Yabiao Wang**[2,3]
**Chengjie Wang**[2]   **Cairong Zhao**[1]*

[1]Tongji University   [2]Tencent YouTu Lab   [3]Zhejiang University   [4]Nanjing University

## Abstract

Multimodal Dataset Distillation (MDD) has emerged as a vital paradigm for enabling efficient training of vision-language models (VLMs) in the era of multimodal data proliferation. Unlike traditional dataset distillation methods that focus on single-modal tasks, MDD presents distinct challenges: (i) the effective distillation of heterogeneous multimodal knowledge, complicated by feature space misalignment and asynchronous optimization dynamics; and (ii) the lack of discrete class guidance, which hinders the distribution coverage and representativeness of synthetic data due to the vastness and continuity of the semantic space. To address these challenges, this paper proposes an Asynchronous Matching with Dynamic sampling (AMD) framework. AMD enables asynchronous trajectory matching by decoupling the selection of starting points for image and text trajectories. Additionally, a Semantics-Aware Prototype Mining module is introduced, which replaces random initialization by leveraging feature-space clustering to identify representative prototypes, enhancing the coverage and representativeness of the distilled samples. Extensive experiments demonstrate that AMD achieves superior distillation performance on Flickr30k and COCO (e.g., IR@1, IR@5, and IR@10 **gains of 4.5%, 9.6%, and 10.9%**, respectively, on Flickr30k 200 pairs.) with negligible computational overhead..

## 1 Introduction

In the era of massive data, the substantial storage, transmission, and computational expenses associated with large-scale datasets pose a significant bottleneck for deep learning model training and iteration. Dataset distillation (DD) (Wang et al., 2018; Zhao et al., 2020; Zhao & Bilen, 2023; Cazenavette et al., 2022) emerged to address this challenge, with the aim of distilling a small amount of synthetic data that allows models trained on this reduced set to achieve performance similar to the original. This approach significantly reduces data volume, lowers training costs, accelerates research, and aids in data privacy protection (Dong et al., 2022; Loo et al., 2023).

However, most existing DD research is mainly focused on single-modal tasks. With the explosion of multimodal data like image-text pairs and the rise of Vision-Language Models (VLMs) (Radford et al., 2021), efficiently processing and utilizing this massive multimodal data presents a new and critical challenge. Consequently, research specifically targeting Multimodal Dataset Distillation (MDD) is becoming exceptionally crucial, offering a vital pathway for efficient multimodal model training and deployment.

Despite its vital role, MDD faces unique challenges distinct from prior dataset distillation paradigms such as image classification (Zhao & Bilen, 2023; Wang et al., 2022; Zhao et al., 2023; Du et al., 2023) and text classification (Li & Li, 2021; Maekawa et al., 2025). MDD presents two main challenges: **(i) Effective distillation of heterogeneous multimodal knowledge.** The core of MDD lies in extracting and condensing effective joint knowledge from heterogeneous modalities like image and text into synthetic data. This process is much more complex than single-modal tasks, as

---

*Corresponding author (`zhaocairong@tongji.edu.cn`). Work done when Ding Qi is an intern in Tencent YouTu Lab, and the advisor is Jian Li.

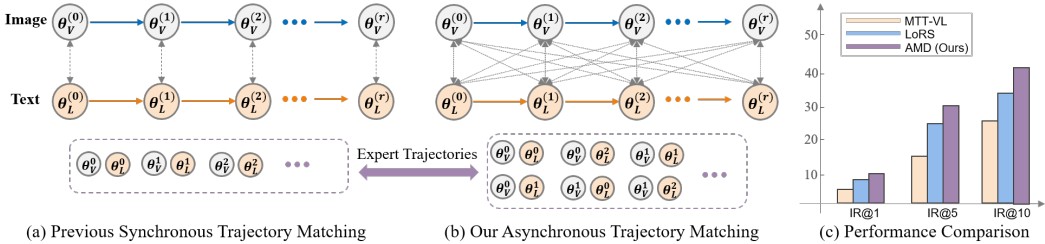

**Figure 1:** Synchronous vs. Asynchronous Matching: (a) Previous synchronous matching only paired same-time expert parameters. (b) Our asynchronous matching explores more flexible and richer cross-modal expert parameter combinations. (c) Performance on Flickr30k: synchronous methods (Wu et al., 2023; Xu et al., 2024) vs. our AMD.

it involves misalignment between modality-specific feature spaces and asynchronous optimization dynamics during training; the latter, in particular, has been largely overlooked in previous work (Wu et al., 2023; Xu et al., 2024) but is crucial for capturing precise cross-modal correlations. **(ii) Distribution coverage and representativeness without discrete class guidance.** Traditional DD methods often benefit from the natural guidance and structure provided by discrete classes. However, MDD lacks such clear class distinctions. Coupled with the vastness and continuity of the image-text data's semantic space, simple random initialization struggles to effectively cover the diverse joint distribution of the original data. Without clear guidance, initial points selected by random methods in previous work may lack representativeness (e.g., corresponding to ambiguous descriptions or low-quality images), affecting distillation quality and subsequent optimization.

To address the aforementioned challenges and limitations, this paper proposes a novel Asynchronous Trajectory Matching (AMD) framework for MDD. In contrast to prior approaches that typically synchronize image and text trajectories by selecting model parameters from the same training stage, as shown in Figure 1(a), our preliminary experiments suggest that such a rigid synchronization is suboptimal for synthesizing modality-specific data representations. This limitation stems from the inherent heterogeneity in the learning dynamics of different modalities. To overcome this, our framework adopts an asynchronous trajectory matching strategy, illustrated in Figure 1(b), that decouples the sampling stages of image and text trajectories, allowing for more diverse combinations of image and text model parameters drawn from different training epochs. This increased flexibility facilitates the optimization of synthetic image-text pairs. Additionally, to address the absence of a discrete set of classes to guide the distillation process, we introduce a Semantics-Aware Prototype Mining module that constructs cluster centers to serve as grounding references. This module performs clustering in the feature space to identify representative sample prototypes. These prototypes replace the randomly selected initial points used in prior methods and are employed to initialize the synthesis process, thereby substantially enhancing the diversity and representativeness of the distilled samples. Notably, these improvements are achieved with negligible additional computational overhead compared to existing methods.

Our main contributions are summarized as follows:

- We propose a novel asynchronous matching with dynamic sampling for MDD that addresses the limitations of synchronous methods by enabling asynchronous sampling of image and text trajectory points to explore richer cross-modal learning dynamics.

- We introduce a semantics-aware prototype mining module that identifies representative prototypes via clustering in the joint semantic space to provide a high-quality initialization, significantly enhancing the coverage and diversity of distilled samples.

- Extensive experiments demonstrate that our method achieves significant performance improvements on Flickr30k and COCO. For instance, on Flickr30k (200 pairs), Image Retrieval metrics IR@1, IR@5, and IR@10 improve by 4.5%, 9.6%, and 10.9%, respectively.

## 2 PRELIMINARY

**Multimodal Dataset Distillation.** We first provide a formal definition of Multimodal Dataset Distillation (MDD). Given a large-scale image-text dataset $\mathcal{T} = \{(x_i, y_i)\}_{i=1}^N$, where $x_i$ denotes an

image sample, $y_i$ represents the corresponding text description, and $N = |\mathcal{T}|$ is the size of the original dataset. The goal of MDD is to compress $\mathcal{T}$ into a budget-constrained synthetic dataset $\mathcal{S} = \{(\tilde{x}_j, \tilde{y}_j)\}_{j=1}^{M}$ with $M \ll N$, such that models trained on $\mathcal{S}$ approximate the performance of those trained on $\mathcal{T}$. This objective can be formulated as:

$$\mathbb{E}_{(x,y)\sim\mathcal{T}_{\text{test}}} \left| \ell(\theta_{\mathcal{V}}^{\mathcal{T}}(x), \theta_{\mathcal{L}}^{\mathcal{T}}(y)) - \ell(\theta_{\mathcal{V}}^{\mathcal{S}}(x), \theta_{\mathcal{L}}^{\mathcal{S}}(y)) \right| \le \epsilon, \tag{1}$$

where $\theta_{\mathcal{V}}^{\mathcal{T}}$ and $\theta_{\mathcal{L}}^{\mathcal{T}}$ are parameters of multimodal model trained on $\mathcal{T}$, $\theta_{\mathcal{V}}^{\mathcal{S}}$ and $\theta_{\mathcal{L}}^{\mathcal{S}}$ are trained on $\mathcal{S}$, $\mathcal{T}_{\text{test}}$ is the test data distribution, $\ell$ denotes the performance measure function, $\epsilon$ is a small tolerance.

Since vision-language datasets lack the category-level labels found in traditional classification tasks, methods such as gradient matching (Zhao et al., 2020) and distribution matching (Zhao & Bilen, 2023), which rely on intra-category data compression, struggle to be effective.

**Matching Image-Text Trajectories.** Existing works (Wu et al., 2023; Xu et al., 2024) employ MTT-based methods (Cazenavette et al., 2022; Cui et al., 2023) to compress key information and cross-modal relationships through: (1) expert trajectories buffering and (2) image-text pairs distilling.

During buffering, the multimodal model is first trained on dataset $\mathcal{T}$ using the bi-directional In-foNCE loss, which consists of symmetric image-to-text and text-to-image contrastive terms. The image-to-text contrastive loss can be formulated as:

$$\mathcal{L}_{\text{InfoNCE}} = -\frac{1}{N} \sum_{i=1}^{N} \log \frac{\exp\left(s(\theta_{\mathcal{V}}(x_i), \theta_{\mathcal{L}}(y_i))/\tau\right)}{\sum_{j=1}^{K} \exp\left(s(\theta_{\mathcal{V}}(x_i), \theta_{\mathcal{L}}(y_j))/\tau\right)}, \tag{2}$$

where $s(\cdot, \cdot)$ measures similarity between positive pair $(x_i, y_i)$ and K negative pairs $(x_i, y_j)$ in a batch, with temperature $\tau$. To ensure the generalization capability of the expert trajectories, it is common practice to perform multiple rounds of retraining and periodically save the parameters of the image encoder $\theta_{\mathcal{V}}$ and the text encoder $\theta_{\mathcal{L}}$ at different training steps, thereby constructing the expert trajectories. One of them can be formalized as: image trajectories $= \{\theta_{\mathcal{V}}^{(0)}, \theta_{\mathcal{V}}^{(1)}, \ldots, \theta_{\mathcal{V}}^{(r)}\}$ and text trajectories $= \{\theta_{\mathcal{L}}^{(0)}, \theta_{\mathcal{L}}^{(1)}, \ldots, \theta_{\mathcal{L}}^{(r)}\}$, r is total training epochs.

During the distilling phase, matching is performed between student and expert trajectories in both the vision ($\mathcal{V}$) and language ($\mathcal{L}$) modalities. At initialization step $t$, the student and expert networks share identical parameters $(\theta_{\mathcal{V}}^{(t)}, \theta_{\mathcal{L}}^{(t)})$. The expert trajectory undergoes $M$ optimization steps to reach $(\theta_{\mathcal{V}}^{(t+M)}, \theta_{\mathcal{L}}^{(t+M)})$, while the student network performs $N$ gradient descent updates ($N \ll M$) to obtain its final parameters $(\tilde{\theta}_{\mathcal{V}}^{(t+N)}, \tilde{\theta}_{\mathcal{L}}^{(t+N)})$. The matching objective minimizes the normalized $\ell_2$-distance between corresponding student and expert trajectories across both modalities:

$$(\tilde{x}, \tilde{y}) = \arg\min_{\tilde{x},\tilde{y}} \left( \frac{\|\tilde{\theta}_{\mathcal{V}}^{(t+N)} - \theta_{\mathcal{V}}^{(t+M)}\|^2}{\|\theta_{\mathcal{V}}^{(t)} - \theta_{\mathcal{V}}^{(t+M)}\|^2} + \frac{\|\tilde{\theta}_{\mathcal{L}}^{(t+N)} - \theta_{\mathcal{L}}^{(t+M)}\|^2}{\|\theta_{\mathcal{L}}^{(t)} - \theta_{\mathcal{L}}^{(t+M)}\|^2} \right), \tag{3}$$

where $(\theta_{\mathcal{V}}^{(t)}, \theta_{\mathcal{L}}^{(t)}) = (\tilde{\theta}_{\mathcal{V}}^{(t)}, \tilde{\theta}_{\mathcal{L}}^{(t)})$. It is noteworthy that, to optimize memory efficiency, LoRS (Xu et al., 2024) leverages TESLA (Cui et al., 2023) technology, enabling the this framework to be executed on a single GPU.

## 3 METHODOLOGY

### 3.1 EXPLORATION

Existing MDD methods typically adopt a synchronous sampling strategy for trajectory matching, perhaps extending the synchronized processing of image and text data in standard VLM training. However, this paper questions the validity of such a synchronous matching assumption:

First, the inherent architectural differences between the image and text networks lead to asynchronous evolution of their parameter trajectories. Taking NFNet+BERT, a commonly used backbone in MDD, as an example, the visual encoder and the text encoder (with BERT often frozen and followed by a linear layer for optimization) exhibit significantly different parameter update dynamics. Second, from the perspective of data distillation, the optimization spaces of synthesized images

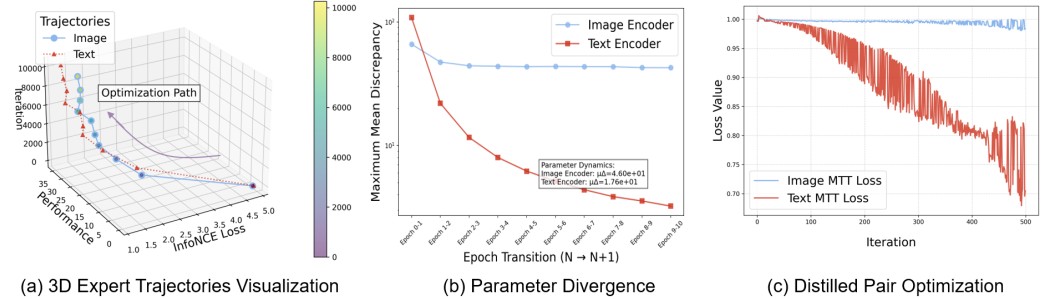

(a) 3D Expert Trajectories Visualization     (b) Parameter Divergence     (c) Distilled Pair Optimization

**Figure 2:** Exploring the Asynchronous Phenomenon: (a) Visualizes the expert trajectories during the buffering stage; (b) analyzes the parameter update magnitudes; (c) MTT loss curves during the distillation stage.

(3×224×224 pixel space) and synthesized texts (768-dimensional embedding space) possess fundamentally different topological properties, making synchronization of their optimization processes difficult.

To empirically investigate these theoretical reasons and validate our skepticism, we conducted systematic experimental analyses on the Flickr30k dataset. Our observations provide strong evidence:

**Observation 1.** Asynchronous Expert Trajectories. Consistent with the notion of inherent architectural differences, our analysis of expert model training reveals clear asynchrony. As visualized in Figure 2 (a), expert trajectories during the buffering stage show initial synchronization followed by clear decoupling in middle and later stages. Furthermore, analysis of parameter update magnitudes (Figure 2 (b)) reveals distinct dynamics: the text network undergoes intense initial fluctuations but quickly converges, whereas the image network maintains a consistently high update intensity throughout training. This evidence highlights the asynchronous evolution of image and text network parameters.

**Observation 2.** Asynchronous Synthetic Data Optimization Speed. Supporting the perspective that the optimization spaces of synthesized images and texts are fundamentally different, Figure 2 (c) demonstrates that during the distillation stage, the synthesized text optimizes significantly faster than the synthesized image. This discrepancy in optimization speed provides direct evidence that the optimization processes of synthesized image and text data are fundamentally asynchronous.

Based on these empirical findings, we draw two important conclusions:

- The asynchrony of expert trajectories is an inherent characteristic of visual-language models;

- Synthetic text converges significantly faster than synthetic images, further validating the asymmetry in the cross-modal optimization process.

Building on these insights, in Section 3.2, we further propose asynchronous trajectory matching — decoupling the distillation paths of image and text modalities to realize a distillation process that better aligns with the actual optimization dynamics.

## 3.2 ASYNCHRONOUS MATCHING WITH DYNAMIC SAMPLING.

Building upon the empirical findings presented in the previous section, which revealed the fundamental asynchronous nature of both expert trajectories and the optimization of synthesized data, we propose a novel **Asynchronous Matching with Dynamic sampling (AMD)** Framework for multimodal Dataset Distillation. Unlike conventional synchronous methods that ignore this inherent asynchrony by strictly aligning expert trajectories based on the same training steps ($t_v = t_l$), AMD, as shown in Figure 3, enables the independent and flexible selection of starting points ($t_v, t_l$) for the image ($\theta_{\mathcal{V}}^{(t)}$) and text ($\theta_{\mathcal{L}}^{(t)}$) expert trajectories (where $t$ is the training step).

The expert and student trajectories whose states are matched in AMD are generated by the standard visual-language model training process using a contrastive loss like InfoNCE. The expert trajectory

is generated by training the network on the real dataset $\mathcal{T}$:

$$(\theta_{\mathcal{V}}^{(t+1)}, \theta_{\mathcal{L}}^{(t+1)}) = (\theta_{\mathcal{V}}^{(t)}, \theta_{\mathcal{L}}^{(t)}) - \alpha_{\mathcal{T}} \nabla \mathcal{L}_{\text{InfoNCE}}(\mathcal{T}; \theta_{\mathcal{V}}^{(t)}, \theta_{\mathcal{L}}^{(t)}) \tag{4}$$

Similarly, the student trajectory corresponds to training a network on the synthetic dataset $\mathcal{S}$:

$$(\tilde{\theta}_{\mathcal{V}}^{(t+1)}, \tilde{\theta}_{\mathcal{L}}^{(t+1)}) = (\tilde{\theta}_{\mathcal{V}}^{(t)}, \tilde{\theta}_{\mathcal{L}}^{(t)}) - \alpha_{\mathcal{S}} \nabla \mathcal{L}_{\text{InfoNCE}}(\mathcal{S}; \tilde{\theta}_{\mathcal{V}}^{(t)}, \tilde{\theta}_{\mathcal{L}}^{(t)}). \tag{5}$$

**Asynchronous Trajectories Matching.** AMD then aims to optimize the synthetic data $\mathcal{S}$ such that points on its resulting student trajectory match points on the expert trajectory. Specifically, it minimizes the asynchronous trajectory matching loss ($L_{AMD}$), which compares student parameters after $N$ steps of optimization on $(\tilde{x}, \tilde{y})$ (corresponding to an expert state) to expert parameters after $M$ steps, starting from $t_v$ and $t_l$. Extending the standard MTT-VL (Wu et al., 2023) formulation, this objective minimizes the normalized $L_2$ distance between student parameters and expert parameters:

$$L_{AMD} = \frac{\|\tilde{\theta}_{\mathcal{V}}^{(t_v+N)} - \theta_{\mathcal{V}}^{(t_v+M)}\|^2}{\|\theta_{\mathcal{V}}^{(t_v)} - \theta_{\mathcal{V}}^{(t_v+M)}\|^2} + \frac{\|\tilde{\theta}_{\mathcal{L}}^{(t_l+N)} - \theta_{\mathcal{L}}^{(t_l+M)}\|^2}{\|\theta_{\mathcal{L}}^{(t_l)} - \theta_{\mathcal{L}}^{(t_l+M)}\|^2} \tag{6}$$
$$\text{s.t.} \quad t_v \in [0, R_V], \ t_l \in [0, R_L],$$

where $R_V$ and $R_L$ denote the sampling ranges for visual and text.

**Maximum Mean Discrepancy based Dynamic Sampling.** We leverage the convergence speed differences between the visual and text expert trajectories to determine dynamic sampling ranges. As shown in Figure 2 (b), we first compute the Maximum Mean Discrepancy (MMD) of trajectory parameters between consecutive epochs. Under a linear kernel, the MMD reduces to the squared Euclidean distance between the average parameter vectors of consecutive epochs. For the visual modality, we define:

$$\mathcal{MMD}_{\mathcal{V},t} = \mathcal{MMD}(\theta_{\mathcal{V}}^{(t-1)}, \theta_{\mathcal{V}}^{(t)}) = \left\| \frac{1}{n_{\mathcal{V}}} \sum_{i=1}^{n_{\mathcal{V}}} \theta_{\mathcal{V},i}^{(t-1)} - \frac{1}{n_{\mathcal{V}}} \sum_{i=1}^{n_{\mathcal{V}}} \theta_{\mathcal{V},i}^{(t)} \right\|^2. \tag{7}$$

Similarly, $\mathcal{MMD}_{\mathcal{L},t}$ is computed for the text modality. In practice, this quantity reflects the update magnitude between consecutive epochs and serves as a surrogate measure of optimization dynamics.

We then compute the median of the trajectory ratio over the complete stored expert trajectory:

$$\mathcal{T}_{\text{median}} = \text{Median}\left( \left\{ \frac{\mathcal{MMD}_{\mathcal{V},t}}{\mathcal{MMD}_{\mathcal{L},t} + \epsilon} \mid t \in [1, T] \right\} \right), \tag{8}$$

where $\epsilon$ is a small constant for numerical stability. The visual and text sampling ranges $R_V$ and $R_L$ are then determined according to the relative position of the trajectory ratio with respect to $\mathcal{T}_{\text{median}}$:

$$R_V = \min\left\{ t \mid \frac{\mathcal{MMD}_{\mathcal{V},t}}{\mathcal{MMD}_{\mathcal{L},t} + \epsilon} > \mathcal{T}_{\text{median}} \right\},$$
$$R_L = \max\left\{ t \mid \frac{\mathcal{MMD}_{\mathcal{V},t}}{\mathcal{MMD}_{\mathcal{L},t} + \epsilon} \leq \mathcal{T}_{\text{median}} \right\}. \tag{9}$$

Intuitively, once the ratio exceeds the median, the text modality has largely stabilized. We therefore truncate text sampling before the crossing point, while allowing visual sampling to extend beyond it. This asymmetric strategy aligns sampling with modality-specific convergence speeds, reducing cross-modal asynchronicity without extra hyperparameters and improving AMD stability and generalization.

### 3.3 SEMANTIC-AWARE PROTOTYPE MINING

Multimodal Dataset Distillation for non-categorical data like image-text pairs faces the critical challenge of insufficient synthetic data coverage and diversity. Due to the data's continuous and complex

**Algorithm 1** Asynchronous Matching with Dynamic Sampling

---

**Input:** Real set $\mathcal{T}$; Expert Trajectories $\{\theta_{\mathcal{V}}^{(t)}\}_{t=0}^{r}, \{\theta_{\mathcal{L}}^{(t)}\}_{t=0}^{r}$; Max start epochs $R_L, R_V$; Synthetic data learning rate $\eta_S$.

**Output:** Synthetic dataset $\mathcal{S}$

1: Initialize $\mathcal{S} = (\tilde{x}, \tilde{y})$ using Semantics-aware Prototype Mining
2: **repeat**
3:     Select the start epoch separately: $t_v \in [0, R_V]$ and $t_l \in [0, R_L]$
4:     Get expert trajectories $\{\theta_{\mathcal{V}}^{(t)}\}_{t=t_v}^{t_v+M}$ and $\{\theta_{\mathcal{L}}^{(t)}\}_{t=l_v}^{l_v+M}$
5:     Initialize the student network $(\theta_{\mathcal{V}}^{(t_v)}, \theta_{\mathcal{L}}^{(t_l)})$
6:     Train $(\theta_{\mathcal{V}}^{(t_v)}, \theta_{\mathcal{V}}^{(t_l)})$ on $\mathcal{S}$ for N steps and get $\{\tilde{\theta}_{\mathcal{V}}^{(t)}\}_{t=t_v}^{t_v+N}$ and $\{\tilde{\theta}_{\mathcal{L}}^{(t)}\}_{t=l_v}^{l_v+N}$
7:     Compute AMD loss using Eq. 6
8:     Update synthetic image-text pairs: $(\tilde{x}, \tilde{y}) \leftarrow (\tilde{x}, \tilde{y}) - \eta_S \nabla_{(\tilde{x}, \tilde{y})} L_{AMD}$
9: **until** convergence
10: **return** $\mathcal{S}$

---

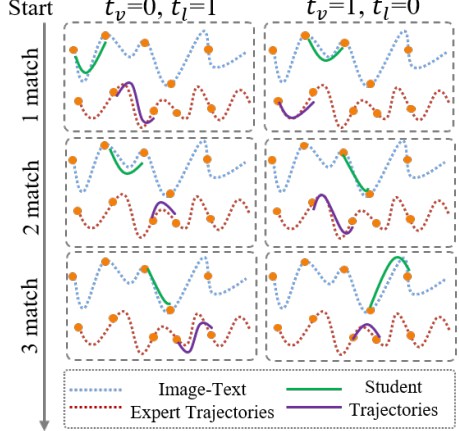

**Figure 3:** Illustration of Asynchronous Trajectory Matching: Dashed lines represent expert trajectories saved during buffering, while solid lines are student trajectories matched during distillation. Columns represent decoupled expert starting points ($(t_v = 0, t_l = 1)$ left, $(t_v = 1, t_l = 0)$ right), and rows depict subsequent matching steps.

nature, traditional random initialization of synthetic samples often selects semantically redundant pairs, significantly harming coverage and diversity. To mitigate this, we propose a novel **Semantics-aware Prototype Mining (SPM)** module. SPM analyzes the joint semantic feature space of the original dataset to identify representative prototypes, which are utilized as initialization points for the $B$ synthetic samples $\{(\tilde{x}_k, \tilde{y}_k)\}_{k=1}^{B}$.

SPM involves several steps. First, we extract corresponding visual features $v_i = \theta_{\mathcal{V}}(x_i)$ and text features $l_i = \theta_{\mathcal{L}}(y_i)$ for every image-text pair $(x_i, y_i)$ in the original dataset $\mathcal{D} = \{(x_i, y_i)\}_{i=1}^{|\mathcal{D}|}$ using trained image and text encoders $\theta_{\mathcal{V}}$ and $\theta_{\mathcal{L}}$. Second, to capture the joint semantic information of each pair, we construct a joint image-text feature $f_i$ by simple feature concatenation: $f_i = [v_i; l_i]$. This builds a representation $\{f_i\}_{i=1}^{|\mathcal{D}|}$ of the original dataset in the joint feature space. Next, we perform K-means clustering on $\{f_i\}$ in this space, setting the number of clusters $K$ equal to the synthetic dataset budget $B$ ($K = B$). This yields $B$ cluster centroids $\{c_k\}_{k=1}^{B}$, each representing a semantic prototype from the original data distribution:

$$\{c_k\}_{k=1}^{B} = \mathcal{C}(\{f_i\}_{i=1}^{|\mathcal{D}|}, \ K = B), (x_k^*, y_k^*) = \arg\min_{x_i, y_i}\|f_i - c_k\|^2 \quad \forall k \in \{1, \ldots, B\}. \tag{10}$$

For each cluster $k \in \{1, \ldots, B\}$, we select the original dataset sample $(x_k^*, y_k^*)$ whose joint feature $f_k^*$ is closest to the centroid $c_k$ as the representative prototype for that cluster.

This prototype-based approach leverages the semantic structure of the original data to guide the initialization process. By selecting $B$ initialization pairs $\{(x_k^*, y_k^*)\}_{k=1}^{B}$ based on diverse semantic clusters identified in the joint feature space, it maximizes the coverage of the original data distribution and ensures the initial synthetic samples are highly diverse and representative. This directly addresses the semantic redundancy issue associated with random initialization, providing a high-quality starting set for the subsequent distillation optimization.

Table 1: Performance comparison with four coreset selection and current multimodal dataset distillation methods on Flickr30k (Plummer et al., 2015) dataset. In line with the fair comparison setting of LoRS (Xu et al., 2024), both LoRS and AMD use 99, 199, and 499 pairs, while others uses 100, 200, and 500 pairs. The NFNet (Brock et al., 2021)+BERT (Vaswani et al., 2017) model trained on the full dataset yields: IR@1=27.3, IR@5=57.1, IR@10=69.7 for I2T, and TR@1=33.9, TR@5=65.1, TR@10=75.2 for T2I. Best results are in bold.

| Pairs | Ratio | Metric | Coreset Selection | | | | Dataset Distillation | | |
|---|---|---|---|---|---|---|---|---|---|
| | | | Rand | Herd | K-Cent | Forget | MTT-VL | LoRS | AMD (Ours) |
| 100 (99) | 0.3% | IR@1 | 1.0 | 0.7 | 0.7 | 0.7 | 4.7±0.2 | 8.3±0.2 | **10.4±0.3** |
| | | IR@5 | 4.0 | 2.8 | 3.1 | 2.4 | 15.7±0.5 | 24.1±0.2 | **30.5±0.7** |
| | | IR@10 | 6.5 | 5.3 | 6.1 | 5.6 | 24.6±1.0 | 35.1±0.3 | **43.0±0.6** |
| | | TR@1 | 1.3 | 1.1 | 0.6 | 1.2 | 9.9±0.3 | 11.8±0.2 | **14.4±0.5** |
| | | TR@5 | 5.9 | 4.7 | 5.0 | 4.2 | 28.3±0.5 | 35.8±0.6 | **39.1±0.6** |
| | | TR@10 | 10.1 | 7.9 | 7.6 | 9.7 | 39.1±0.7 | 49.2±0.5 | **52.6±0.6** |
| 200 (199) | 0.7% | IR@1 | 1.1 | 1.5 | 1.5 | 1.2 | 4.6±0.9 | 8.6±0.3 | **13.1±0.3** |
| | | IR@5 | 4.8 | 5.5 | 5.4 | 3.1 | 16.0±1.6 | 25.3±0.2 | **34.9±0.6** |
| | | IR@10 | 9.2 | 9.3 | 9.9 | 8.4 | 25.5±2.6 | 36.6±0.3 | **47.5±0.7** |
| | | TR@1 | 2.1 | 2.3 | 2.2 | 1.5 | 10.2±0.8 | 14.5±0.5 | **16.9±0.4** |
| | | TR@5 | 8.7 | 8.4 | 8.2 | 8.4 | 28.7±1.0 | 38.7±0.5 | **42.3±0.6** |
| | | TR@10 | 13.2 | 14.4 | 13.5 | 10.2 | 41.9±1.9 | 53.4±0.5 | **56.2±0.8** |
| 500 (499) | 1.7% | IR@1 | 2.4 | 3.0 | 3.5 | 1.8 | 6.6±0.3 | 10.0±0.2 | **15.8±0.4** |
| | | IR@5 | 10.5 | 10.0 | 10.4 | 9.0 | 20.2±1.2 | 28.9±0.7 | **39.8±0.4** |
| | | IR@10 | 17.4 | 17.0 | 17.3 | 15.9 | 30.0±2.1 | 41.6±0.6 | **53.2±0.5** |
| | | TR@1 | 5.2 | 5.1 | 4.9 | 3.6 | 13.3±0.6 | 15.5±0.7 | **19.3±0.5** |
| | | TR@5 | 18.3 | 16.4 | 16.4 | 12.3 | 32.8±1.8 | 39.8±0.4 | **46.4±0.4** |
| | | TR@10 | 25.7 | 24.3 | 23.3 | 19.3 | 46.8±0.8 | 53.7±0.3 | **60.0±0.6** |

# 4 EXPERIMENTS

## 4.1 EXPERIMENTS SETUP

**Datasets and Metrics.** Following established MDD protocols (Wu et al., 2023; Xu et al., 2024), we evaluate our approach on the Flickr30K (Plummer et al., 2015) and COCO (Lin et al., 2014) datasets, which are standard cross-modal retrieval benchmarks containing 31,783 and 123,287 images, respectively, each paired with five human-annotated captions. We assess retrieval performance using standard Recall@K (R@K) metrics with $K \in \{1, 5, 10\}$, reporting results in both directions: Image-to-Text (I2T), denoted as IR@K, which measures the hit rate of retrieving correct captions among the top-K results, and Text-to-Image (T2I), denoted as TR@K, which evaluates the accuracy of finding matching images based on text queries.

**Implementation Details.** Following the setup of the LoRS (Xu et al., 2024) baseline, we utilize an NFNet (Brock et al., 2021) (Normalizer-Free ResNet) pretrained on ImageNet (Deng et al., 2009) as the image encoder, along with a pretrained BERT (Vaswani et al., 2017) model that includes an appended linear layer as the text encoder. In accordance with the protocols of previous work (Xu et al., 2024; Wu et al., 2023), the BERT weights remain frozen during training and distillation, with only the parameters of the linear layer being optimized. Adhering to the MTT (Cazenavette et al., 2022), during the buffer phase, we train the image and text encoders on the original dataset for 10 epochs, repeating this process 20 times to generate 20 expert trajectories. We optimize the distilled data using SGD with momentum 0.5. The reported results are calculated as the mean ± standard deviation over 15 independent evaluations: we generate 3 synthetic datasets, and for each dataset, we retrain the model 5 times. All experiments are conducted on a single NVIDIA V100 / RTX 4090 GPU. More detailed are provided in the Appendix.

**Counterpart Methods.** We compared two main categories of methods: Coreset Selection and Dataset Distillation. Coreset Selection includes commonly used techniques such as Random (Rebuffi et al., 2017), Herding (Welling, 2009), K-center (Farahani & Hekmatfar, 2009), and Forgetting (Toneva et al., 2018). Dataset Distillation encompasses MTT-VL (Wu et al., 2023) and LoRS (Xu et al., 2024). MTT-VL is the first work to apply MTT (Cazenavette et al., 2022) (training trajectory matching) in the multimodal area, while LoRS enhances similarity mining and incorporates TESLA (Cui et al., 2023) technology to reduce memory overhead.

Table 2: Performance comparison with coreset selection and dataset distillation methods on COCO (Lin et al., 2014). For a fair comparison, LoRS (Xu et al., 2024) and AMD use 99, 199, and 499 pairs, while others use 100, 200, and 500 pairs. The NFNet (Brock et al., 2021)+BERT (Vaswani et al., 2017) model trained on the full dataset achieves: IR@1=16.9, IR@5=41.9, IR@10=55.9 for I2T, and TR@1=19.6, TR@5=45.6, TR@10=59.5 for T2I. Best results are in bold.

| Pairs | Ratio | Metric | Coreset Selection | | | | Dataset Distillation | | |
|---|---|---|---|---|---|---|---|---|---|
| | | | Rand | Herd | K-Cent | Forget | MTT-VL | LoRS | AMD (Ours) |
| 100 (99) | 0.02% | IR@1 | 0.3 | 0.5 | 0.4 | 0.3 | 1.3±0.1 | 1.8±0.1 | **2.8±0.2** |
| | | IR@5 | 1.3 | 1.4 | 1.4 | 1.5 | 5.4±0.3 | 7.1±0.2 | **10.5±0.2** |
| | | IR@10 | 2.7 | 3.5 | 2.5 | 2.5 | 9.5±0.5 | 12.2±0.2 | **17.2±0.4** |
| | | TR@1 | 0.8 | 0.8 | 1.4 | 0.7 | 2.5±0.3 | 3.3±0.2 | **4.1±0.3** |
| | | TR@5 | 3.0 | 2.1 | 3.7 | 2.6 | 10.0±0.5 | 12.2±0.3 | **13.8±0.3** |
| | | TR@10 | 5.0 | 4.9 | 5.5 | 4.8 | 15.7±0.4 | 19.6±0.3 | **21.8±0.4** |
| 200 (199) | 0.04% | IR@1 | 0.6 | 0.9 | 0.7 | 0.6 | 1.7±0.1 | 2.4±0.1 | **3.8±0.2** |
| | | IR@5 | 2.3 | 2.4 | 2.1 | 2.8 | 6.5±0.4 | 9.3±0.2 | **13.4±0.3** |
| | | IR@10 | 4.4 | 4.1 | 5.8 | 4.9 | 12.3±0.8 | 15.5±0.2 | **21.4±0.4** |
| | | TR@1 | 1.0 | 1.0 | 1.2 | 1.1 | 3.3±0.2 | 4.3±0.1 | **4.6±0.2** |
| | | TR@5 | 4.0 | 3.6 | 3.8 | 3.5 | 11.9±0.6 | 14.2±0.3 | **15.5±0.6** |
| | | TR@10 | 7.2 | 7.7 | 7.5 | 7.0 | 19.4±1.2 | 22.6±0.2 | **24.1±0.5** |
| 500 (499) | 0.09% | IR@1 | 1.1 | 1.7 | 1.1 | 0.8 | 2.5±0.5 | 2.8±0.2 | **4.2±0.2** |
| | | IR@5 | 5.0 | 5.3 | 6.3 | 5.8 | 8.9±0.7 | 9.9±0.5 | **14.2±0.5** |
| | | IR@10 | 8.7 | 9.9 | 10.5 | 8.2 | 15.8±1.5 | 16.5±0.7 | **22.3±0.4** |
| | | TR@1 | 1.9 | 1.9 | 2.5 | 2.1 | 5.0±0.4 | 5.3±0.5 | **5.7±0.6** |
| | | TR@5 | 7.5 | 7.8 | 8.7 | 8.2 | 17.2±1.3 | 18.3±1.5 | **19.3±1.2** |
| | | TR@10 | 12.5 | 13.7 | 14.3 | 13.0 | 26.0±1.9 | 27.9±1.4 | **28.7±1.0** |

## 4.2 QUANTITATIVE RESULTS

As shown in Tables 1 and 2, our AMD demonstrates significant advantages over existing approaches across both datasets. On the Flickr30k dataset, AMD achieves a new state-of-the-art performance, significantly surpassing traditional Coreset Selection methods and existing dataset distillation techniques. For instance, under the setting of 200 image-text pairs, AMD improves I2T over the LoRS baseline with gains of +4.5%, +9.6%, and +10.9% in IR@1, IR@5, and IR@10, respectively. For T2I, AMD achieves improvements of +2.4%, +3.6%, and +2.8% in TR@1, TR@5, and TR@10, respectively. Given that the COCO dataset is 3.9x larger than Flickr30k and contains richer semantic relationships, the performance for all methods are relatively lower. Nevertheless, AMD maintains its superiority on the more complex COCO dataset. For example, under the setting of 200 pairs, AMD achieves improvements of +1.4%, +4.1%, and +5.9% in IR@1, IR@5, and IR@10, respectively, compared to the LoRS baseline.

Additionally, we observed two noteworthy phenomena. First, as the distillation budget (the number of image-text pairs) increases, the performance gains from the asynchronous trajectory become more pronounced. In the case of Flickr30k, AMD outperformed LoRS by 2.1%, 4.5%, and 5.8% in the IR@1 metric for 99, 199, and 499 pairs, respectively. This suggests that previous synchronous trajectory strategies may create performance bottlenecks when handling large-scale data, as they struggle to adapt to the increasing complexity of image-text pairs. Second, the improvements in I2T retrieval metrics are particularly significant, indicating that the previous synchronous trajectory approach, which forced the matching of imbalanced image and text expert trajectories, led to optimization challenges for synthetic image-text pairs. In contrast, our proposed asynchronous trajectory technique enables flexible optimization of image and text combinations at different stages, resulting in substantial performance enhancements.

## 4.3 ABLATION STUDY

**Component Analysis.** We conduct a comprehensive ablation study on the proposed AMD framework to analyze the contribution of its key components: (1) Baseline (using the codebase of LoRS (Xu et al., 2024)), (2) Asynchronous Trajectory Matching (AMD), and (3) Semantic-aware Prototype Mining (SPM). As shown in Table 9, the Baseline (LoRS) provides a solid starting point. Adding the AMD component alone yields significant improvements across all metrics (e.g., IR@1 increases from 8.6% to 12.1%), highlighting the effectiveness of explicitly modeling and leveraging asynchronous learning dynamics during distillation. Adding the SPM component to the baseline also provides noticeable gains (e.g., IR@1 increases from 8.6% to 9.1%), demonstrating the importance of semantic-aware initialization for enhancing synthetic data coverage and diversity. Importantly, the

Table 3: Ablation study of proposed modules. Experiments are conducted on the Flickr30k 200pairs setting. The best results are in bold.

| Baseline | AMD | SPM | IR@1 | IR@5 | IR@10 | TR@1 | TR@5 | TR@10 |
|---|---|---|---|---|---|---|---|---|
| ✓ | | | 8.6±0.3 | 25.3±0.2 | 36.6±0.3 | 14.5±0.5 | 38.7±0.5 | 53.4±0.5 |
| ✓ | ✓ | | 12.1±0.3 | 33.9±0.6 | 46.7±0.5 | 16.5±0.4 | 41.6±0.6 | 55.7±0.5 |
| ✓ | | ✓ | 9.1±0.1 | 26.4±0.3 | 38.5±0.4 | 15.3±0.4 | 40.1±0.5 | 53.9±0.4 |
| ✓ | ✓ | ✓ | **13.1±0.3** | **34.9±0.6** | **47.5±0.7** | **16.9±0.4** | **42.3±0.6** | **56.2±0.8** |

Table 4: Cross-architecture generalization: We utilize NFNet+BERT as the dataset distillation model to generate synthetic images and evaluate performance across various architectures.

| Pairs | Method | Evaluate Model | Flickr30k | | | | | |
|---|---|---|---|---|---|---|---|---|
| | | | IR@1 | IR@5 | IR@10 | TR@1 | TR@5 | TR@10 |
| 499 | LoRS | NFNet+BERT | 10.0±0.2 | 28.9±0.7 | 41.6±0.6 | 15.5±0.7 | 39.8±0.4 | 53.7±0.3 |
| | | ResNet+BERT | 3.3±0.2 | 12.7±0.3 | 20.4±0.2 | 6.8±0.2 | 19.6±1.3 | 31.1±0.3 |
| | | RegNet+BERT | 3.5±0.1 | 12.6±0.3 | 21.1±0.4 | 6.8±0.3 | 20.8±0.3 | 30.2±0.3 |
| 499 | AMD | NFNet+BERT | 15.8±0.4 | 39.1±0.4 | 53.2±0.5 | 19.3±0.5 | 46.4±0.4 | 60.0±0.6 |
| | | ResNet+BERT | 4.1±0.3 | 14.2±0.5 | 22.6±0.4 | 7.6±0.4 | 22.3±0.4 | 33.2±0.6 |
| | | RegNet+BERT | 4.0±0.2 | 14.4±0.3 | 23.1±0.5 | 7.5±0.3 | 22.8±0.5 | 32.7±0.4 |

Table 5: Upper Bound Analysis on Synthetic Data Scale.

| Method | Image Encoder | Text Encoder | Ratio | TR@1 | TR@10 | IR@1 | IR@10 |
|---|---|---|---|---|---|---|---|
| AMD | NFNet | BERT | 10% | 32.5 | 73.9 | 24.7 | 67.3 |
| Upper Bound | NFNet | BERT | 100% | 33.9 | 75.2 | 27.3 | 69.7 |
| AMD | NFNet | CLIP | 10% | 60.5 | 91.7 | 47.9 | 86.8 |
| Upper Bound | NFNet | CLIP | 100% | 61.2 | 92.8 | 49.8 | 88.3 |

a black and gray bucking bronco is attempting to buck off a cowboy at the rodeo | a man in a carhart shirt runs to help the cowboy who fell off of the cow during the rodeo | greyhounds race on a sandy track with the dog in green taking the lead | two horses are right along side each other in the race | a band performs music for a crowd of people | a keyboard player, a drummer, a guitarist and a violinist play their instruments on stage

**Figure 4:** Visualization of initial (Left) and synthetic (Right) image-text pairs, with the synthetic data undergoing 3000 distillation steps.

improvement from AMD alone is generally larger than that from SPM alone, particularly for Image Retrieval metrics. When combining both AMD and SPM, our full framework achieves the best performance across all metrics, reaching **13.1%/34.9%/47.5%** for IR and **16.9%/42.3%/56.2%** for TR. These results underscore the complementarity of the proposed AMD and SPM components, and their combined effect leads to superior performance in MDD.

**Cross-Architecture Generation.** Cross architecture generation aims to assess the ability of synthetic datasets to generalize to unseen architectures. We employed NFNet+BERT as the distillation model to generate the synthetic dataset, and then evaluated it using various architectures. Since BERT is frozen, we followed the baseline approach of LoRS by modifying the image encoder architecture, including ResNet (He et al., 2016) and RegNet (Radosavovic et al., 2020). As shown in Table 4, the dataset generated by our AMD method outperforms the baseline LoRS in cross-architecture performance.

**Upper Bound Analysis.** We investigated the AMD performance upper bound by scaling up synthetic data quantity. As detailed in Table 5, with the CLIP encoder, AMD trained on a 10% synthetic subset achieves 47.9 IR@1, recovering over 96% of the full dataset's upper bound (49.8). The 10% synthetic subset, with the BERT encoder, recovers 90.4% of its upper bound (24.7 vs. 27.3 IR@1). This demonstrates the high performance ceiling and potential of the AMD approach.

## 4.4 QUALITATIVE RESULTS.

**Synthetic Image-Text Pairs Visualization.** Figure 4 provides qualitative examples of synthetic image-text pairs generated by our method. Similar to prior work (Wu et al., 2023; Xu et al., 2024), our synthetic images exhibit a deepdream-like style (Zeiler & Fergus, 2014), characterized by real

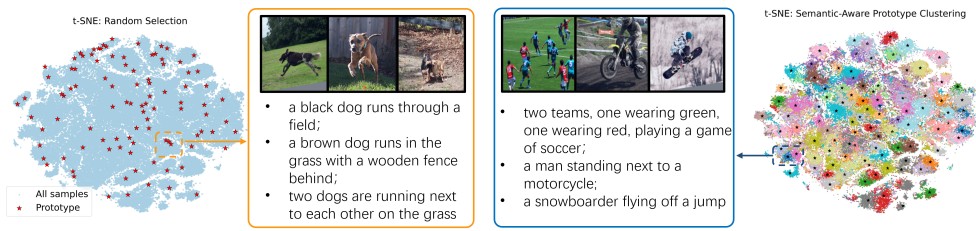

**Figure 5:** t-SNE visualization of initialization strategies. (Left) Random selects samples haphazardly, leading to semantic redundancy and poor coverage. (Right) SPM (ours) identifies representative prototypes via clustering, ensuring uniform, diverse coverage of the semantic manifold. Each cluster is color-coded.

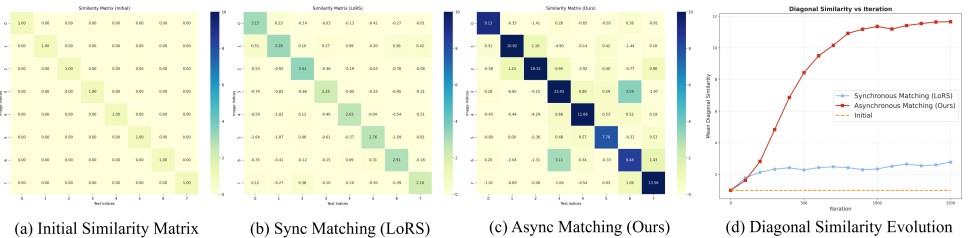

(a) Initial Similarity Matrix  (b) Sync Matching (LoRS)  (c) Async Matching (Ours)  (d) Diagonal Similarity Evolution

**Figure 6:** Qualitative analysis of similarity matrices and their evolution. (a) The initial similarity matrix, where only diagonal elements are 1.0. (b) The similarity matrix after synchronous matching using the LoRS baseline. (c) The similarity matrix after asynchronous matching with our AMD framework. (d) The evolution of mean diagonal similarity over iterations, demonstrating the superior convergence of AMD compared to the LoRS.

images overlaid with learned high-frequency components. The examples demonstrate that our synthetic text, even if not longer, is significantly more effective at capturing salient objects and relationships compared to the initial synthetic text (i.e., before optimization). For instance, the last case clearly describes the performers and their relationships, a detail often lacking initially but crucial for high-quality VL understanding.

**Initialization Strategies Analysis.** As shown in Figure 5, t-SNE visualization reveals that randomly initialized prototypes (red stars) cluster around a few similar semantics (e.g., multiple "dogs running on grass"), leading to limited and biased coverage. In contrast, SPM employs K-means to select representative prototypes that uniformly span distinct semantic clusters, such as "soccer matches," "motocross," and "snowboarding." This diversity-driven initialization improves semantic coverage and fidelity, providing a stronger foundation for subsequent distillation.

**Similarity Matrix Evolution.** As shown in Figure 6, we visualize the evolution of the low-rank similarity matrix during distillation. Figures (a)–(c) correspond to initialization, LoRS, and AMD, respectively. By decoupling image and text trajectories, AMD enables stable text optimization via asynchronous matching at optimal convergence and frees image synthesis from stage-wise constraints, allowing optimization along more informative gradients. This results in stronger diagonal dominance and better suppression of off-diagonal elements. Figure (d) further shows that AMD converges faster and achieves higher final diagonal values than LoRS.

## 5 CONCLUSION

In this study, we introduce the Asynchronous Matching with Dynamic Sampling framework for Multimodal Dataset Distillation, AMD. We propose two novel components: an asynchronous trajectory matching strategy that decouples image and text parameter sampling to leverage differential convergence rates, and a semantics-aware prototype mining module leveraging clustering for representative initialization. Experimental results demonstrate AMD achieves superior distillation performance on Flickr30k and COCO with negligible computational overhead, offering an efficient and scalable solution for mitigating data bottlenecks.

## 6 ACKNOWLEDGMENT

This work was supported by National Natural Science Fund of China (No. U25A20527, 62473286). This work was also supported by Shanghai Municipal Science and Technology Major Project (No. 2025SHZDZX025G10).

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

Table 6: Ablation study on Text Encoder.

| Text Encoder | Method | TR@1 | TR@10 | IR@1 | IR@10 |
|---|---|---|---|---|---|
| BERT | MTT-VL | 9.9 | 39.1 | 4.7 | 24.6 |
| | AMD | 14.4 | 52.6 | 10.4 | 43 |
| CLIP | MTT-VL | 31.4 | 72 | 17.1 | 56.2 |
| | AMD | 41.7 | 81.2 | 29.7 | 76.4 |

# A   APPENDIX

The supplementary material is organized as follows: Section A.1 reviews related works on dataset distillation and multimodal dataset distillation. Section A.2 provides a performance comparison using different text and visual encoders. Section A.3 analyzes the time overhead of the AMD. Section A.4 provides implementation details and hyperparameter settings.

## A.1   RELATED WORK

**Dataset Distillation.** Dataset distillation Wang et al. (2018); Zhao et al. (2020) has witnessed rapid advancements in recent years, with the primary goal of generating a compact set of highly informative synthetic data to replace massive original datasets for model training, thereby dramatically reducing demands on data storage and computational resources. Current research predominantly focuses on classification tasks, giving rise to three mainstream approaches: gradient matching Zhao et al. (2020), distribution matching Zhao & Bilen (2023), and training trajectory matching Cazenavette et al. (2022).

Gradient matching methods (DC) optimize synthetic samples by minimizing the gradient discrepancies between synthetic and real data during model training. Subsequent improvements include DSA's Zhao & Bilen (2021) introduction of differentiable data augmentation to enhance generalization, and IDC's Kim et al. (2022) adoption of multi-formation synthesis techniques for better performance. Distribution matching methods (DM) aim to align the statistical distributions of synthetic data with real data in feature space, where early works employed Maximum Mean Discrepancy (MMD) as the distance metric, while advanced approaches like CAFE Wang et al. (2022) extend the alignment to intermediate network layers beyond final features. Training trajectory matching methods (MTT) optimize synthetic data by minimizing parameter differences between models trained on synthetic versus real data across training stages. To address the prohibitive computational and memory costs of long-horizon trajectory matching, TESLA Cui et al. (2023) reduces memory consumption through loss reparameterization.

**Multimodal Dataset Distillation.** In multimodal dataset distillation, current research primarily targets image-text paired datasets. MTT-VL Wu et al. (2023) pioneered the first framework by extending conventional single-trajectory matching to dual visual-textual trajectory alignment. Building upon this, LoRS Xu et al. (2024) introduces cross-modal similarity mining with low-rank matrices to reduce computational overhead, while incorporating TESLA's memory management techniques to enable efficient single-GPU training.

## A.2   FURTHER ARCHITECTURE EXPERIMENTS

Following the evaluation protocol of MTT-VL Wu et al. (2023), we systematically replaced the text encoder and the image encoder to verify the generalization performance of AMD.

As shown in Table 6, the performance is significantly stronger when the text encoder utilizes CLIP. This is attributed to the fact that pre-trained CLIP inherently possesses a more powerful cross-modal alignment capability and richer semantic information. Furthermore, the performance gain of AMD compared to MTT-VL (e.g., in terms of the IR@1 metric) is more pronounced when using CLIP as the text encoder, which fully validates the effectiveness of our proposed method.

In addition, as demonstrated in Table 7, different visual encoders have a more pronounced impact on the IR performance on synthetic data, where a better visual encoder consistently brings a certain degree of improvement. For instance, the overall performance is superior when using the more advanced ViT as the visual encoder compared to using NFNet, NFResNet and NFRegNet. This ob-

Table 7: Ablation study on Image Encoder.

| Image Encoder | Method | TR@1 | TR@10 | IR@1 | IR@10 |
|---|---|---|---|---|---|
| ViT | MTT-VL | 10.4 | 38.7 | 5.4 | 27.4 |
| | AMD | 14.4 | 52.6 | 10.4 | 43 |
| NFNet | MTT-VL | 9.9 | 39.1 | 4.7 | 24.6 |
| | AMD | 14.1 | 51.9 | 9.7 | 40.8 |
| NFResNet | MTT-VL | 6.5 | 28.1 | 3.5 | 18.7 |
| | AMD | 10.9 | 43.4 | 7.9 | 33.6 |
| NFRegNet | MTT-VL | 7.8 | 33.3 | 3.3 | 20.5 |
| | AMD | 11.8 | 46.2 | 8.6 | 35.7 |

Table 8: Comparison of per-iteration training time between AMD and LoRS.

| Dataset | Method | 100 pairs | 200 pairs | 500 pairs |
|---|---|---|---|---|
| Flickr30k | LoRS | 6.44s | 6.63s | 6.56s |
| | AMD | 6.52s | 6.67s | 6.61s |
| COCO | LoRS | 6.13s | 6.04s | 6.09s |
| | AMD | 6.21s | 6.15s | 6.21s |

Table 9: Hyperparameters for different experiments.

| Dataset | Flickr30k | | | COCO | | |
|---|---|---|---|---|---|---|
| Pairs | 100 | 200 | 500 | 100 | 200 | 500 |
| lr_image | 100 | 1000 | 1000 | 1000 | 1000 | 5000 |
| lr_text | 100 | 1000 | 1000 | 1000 | 1000 | 5000 |
| lr_lr | 0.001 | 0.01 | 0.01 | 0.01 | 0.01 | 0.01 |
| lr_similarity | 10 | 10 | 100 | 5 | 50 | 500 |
| synth steps | 8 | 8 | 8 | 8 | 8 | 8 |

servation suggests that AMD can effectively leverage more powerful visual feature representations to enhance the matching accuracy for synthetic data.

### A.3 TIME COST ANALYSIS

We base our implementation on the LoRS codebase, thus comparing our time cost with LoRS. As shown in Table 8, the per-iteration time costs of AMD and LoRS are highly consistent across datasets and pair counts. Crucially, the average per-iteration time reported for AMD explicitly includes the amortization of the one-time SPM initialization cost (which is only 5-10 minutes). For a complete distillation run (3000 iterations), the total time is approximately 5.4 hours (e.g., 6.52seconds $\times$ 3000 $\approx$ 5.43 hours for Flickr30k 100 pairs), which is consistent with the baseline. This confirms our significant performance improvements are achieved with virtually no additional computational cost during distillation.

### A.4 MORE IMPLEMENTATION DETAILS

We've released the hyperparameter configuration for AMD, which is aligned with the baseline LoRS Xu et al. (2024).

### A.5 MORE VISUALIZATION

We provide more visual comparisons of synthesized images and text before and after distillation, as shown in Figure 7-10.

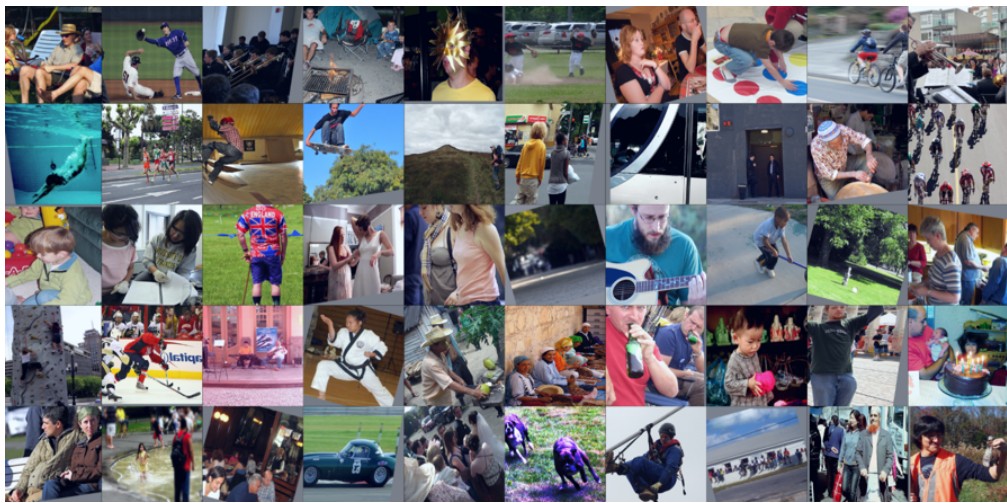

**Figure 7:** Visualization for Initialized images, interation=0.

a group of friends relaxing and enjoying a nice sunny day in the park
a texas baseball player catches the ball as the 12 player from the opposite team slides into base with an umpire standing by, ready to make the call
a orchestra performing a piece
in front of two tents, a man and two children sit near a campfire
a man wears a gold sun mask
kids playing ball in the park
a woman and a man sitting at a table eating
two young boys in casual attire playing twister with other children
two people are riding bicycles along a road
a man in black clothing plays a trumpet
a woman in a bathing suit is diving into a pool
runners at a marathon running a race heading for the finish line
a couple of young kids skateboarding on ramps
a man jumps in the air on his skateboard
a man with a big backpack is walking through a grass trail up a hill
a woman carrying a bag standing next to a man waiting to cross the street
a woman in a blue top is sitting on a bus
two men standing outside next to a building
a man is carving an object out of clay
a bunch of cyclist are riding their bikes down the road
two little boys are playing with toys
two women looking at information in a spiral booklet
a man with a cane is standing on the grass
a woman is helping another woman with the closure on her dress
two women are walking together
a little boy in green pants and a white shirt is standing in the street
a man with a beard and a blue shirt plays a guitar
a boy playing with a wheel on a stick
a young child playing with an empty bucket in the grass
two men and two women are preparing a large meal in the kitchen
two people are climbing a portable rock wall
two opposing hockey players make a play for the puck with opposing fans and team members watching
the musicians are playing
young women practicing marshal arts in a gym
a man cutting open a fruit with a large knife
women are working with baskets of food
two men are drinking beer
a toddler holding a pink piece of yarn
a man holds up a free hugs sign above his head
a little girl looks at a birthday cake while a man holds a baby
a man and woman sit on a park bench
a little girl dressed in yellow splashes in a shallow pool
people are enjoying food at a crowded restaurant
a green sports car with the number 63 is driving on a track
a bride and groom outside with their guests
two brindle dogs running in the grass
a man in a blue suit and brown boots is hanging on a harness on a metal pole
a group of children are sitting on a wall
a man with a red beard pushes a cart along a sidewalk
a man wearing an orange safety vest is holding a rifle

**Figure 8:** Visualization for Initialized texts, interation=0

# B   THE USE OF LARGE LANGUAGE MODELS (LLMS)

We use LLMs solely for checking grammar and polishing writing.

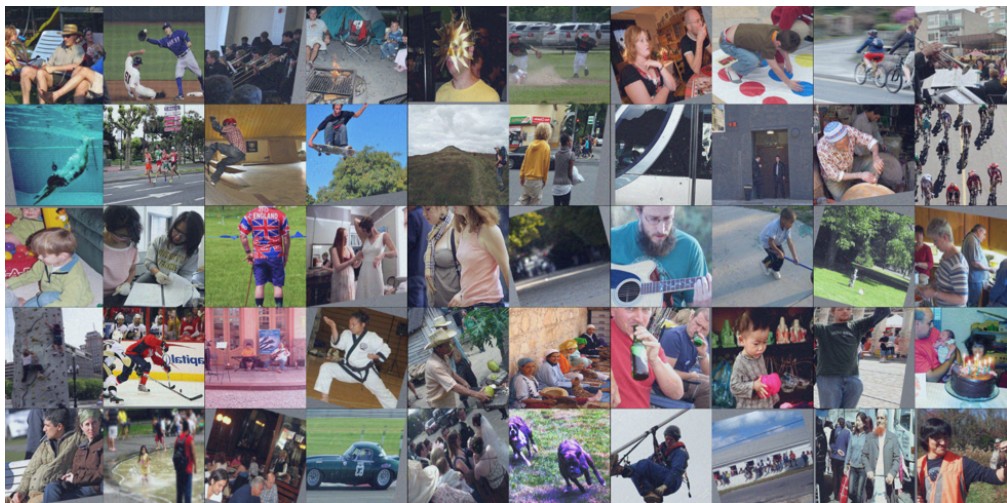

**Figure 9:** Visualization for distilled images, interation=3000.

a group of friends relaxing and enjoying a nice sunny day in the park
pitcher caught just after his pitch, body still in pose
guitarist and drummer on stage at a concert
grilling ribs, chicken kabobs, and vegetable kabobs
contortionist in strange checkered outfit wearing a white mask
little girl kicking rocks on the beach with her dog
women sitting on a couch drinking beer
a child wearing blue tee-shirt playing with an orange pinata without a blindfold
several men are playing ice hockey in an arena
an orchestra performs
boy takes a bath with diving mask and snorkel
runners at a marathon running a race heading for the finish line
dogs running and playing in a grassy area
a man jumps in the air on his skateboard
old couple walking through a field
young white male child with blond-hair in a red shirt coloring with crayons outside with an adult
a girl with dark hair is gazing out the window of a train car
two men standing outside of a brick building
small children in a third world country sitting together
many people out on the street on a clear day riding bikes and walking
mother and daughter playing a board game
two women looking at information in a spiral booklet
a smiling man on a horse in front of brush and woods
a bride is being helped into her white wedding dress by one of her bridesmaids, who is wearing a red dress
dog watches woman eating alone
a child wearing a brown coat, red hat and snow boots on top of a snow pile near a tree on the corner of a street intersection
asian schoolgirl carrying her bags and a musical instrument
kid playing near water fountain
children playing ball in a green field on a sunny day
2 ladies one has her hands on her hips smiling and the other one is holding something up with her other hand behind her
climber climbing an ice wall
two green bay packers hi-five to celebrate a touchdown
their are three women at a desk and the women with the long braid looks in the microscope
a kickboxer practicing on the heavy bag
middle-aged hispanic woman sweeping a sidewalk
black women make cloths in their home
female police officers wearing sunglasses
mother holding newborn infant between her grandparents while sitting on a couch
a group of people at the beach jumping in the air simultaneously for the camera
a woman holds a baby while lying on a couch
a man and woman sit on a park bench
a child in a blue shirt running under a fountain
asian man drinking at a booth in a restaurant
formula one cars racing which the red car seems to be winning
a bride and groom cutting the cake at their wedding
a running greyhound
a man in dark colored clothing skies over ledge, hanging in midair
tourists walking a german side street where souvenirs are sold
a woman showing a small dog to an infant
group of people outside in the city videotaping a show

**Figure 10:** Visualization for distilled texts, interation=3000.

