# OpenReview forum: "Asynchronous Matching with Dynamic Sampling for Multimodal Dataset Distillation"
_ICLR.cc/2026/Conference — ICLR 2026 Poster_

### Official Review · Reviewer_EB68 · 2025-10-18

**Soundness:** 3
**Presentation:** 2
**Contribution:** 1
**Rating:** 4
**Confidence:** 3

**Summary:**

This paper focuses on Multimodal Dataset Distillation (MDD), a task that aims to distill a small amount of synthetic data capable of achieving performance comparable to the original dataset. To address the challenge of discrepancy between image and text distillation stages, this work proposes an asynchronous matching strategy. Furthermore, it introduces the Semantics-aware Prototype Mining (SPM) module for initializing the searching space and identifying representative prototypes. Experimental results on the Flickr30K and COCO datasets demonstrate the effectiveness of the proposed AMD method.

**Strengths:**

1. The motivation is strong, underpinned by a solid experimental analysis (Sec 3.1) that effectively highlights the discrepancy problem in previous multimodal distillation methods. This makes the proposed asynchronous method (AMD) well-justified.

2. The proposed algorithm is commendably simple yet effective. The substantial improvements over prior LoRS work (Tables 1 & 2) clearly demonstrate the effectiveness and superiority of the asynchronous matching strategy.

**Weaknesses:**

1. My primary concern is the sufficiency of the contribution. While the paper identifies its core contribution as asynchronous matching with dynamic sampling for MDD, the algorithm appears to be summarized as matching teacher and student networks within controlled ranges $R_L$ and $R_V$. Could the authors clarify if this understanding is accurate, or if I am oversimplifying the approach?

2. Regarding the Semantic Prototype Mining (SPM) module, it appears to function more as an initialization trick than a core MDD-specific method. SPM uses K-means to identify $B$ representative prototypes based on concatenated features, which then serves as an initialization for the subsequent matching process. Critically, this module seems unrelated to the 'asynchronous' matching strategy. Moreover, it comes across as an effective heuristic rather than a genuinely learnable component.

3. The authors claim (L86) that the achieved improvements incur negligible additional computational overhead. However, this assertion lacks supporting evidence in the main paper. While Appendix Table 8 compares running times, it notably omits the computational cost of SPM initialization, specifically the K-means clustering time.

**Questions:**

1. I am confused by the statement in L198 regarding "decoupling the distillation paths of images and text modalities." From Figure 1, the proposed asynchronous matching appears to couple the two modality trajectories at different timestamps. In contrast, previous synchronous methods seem to operate in a more decoupled manner, as image and text trajectories are matched strictly pair by pair.

2. What specific value is chosen for $B$ (the number of clusters) in the SPM module? Furthermore, are there any ablation studies investigating the impact of different $B$ values?

3. Could the authors provide a more detailed explanation of Figure 3? Specifically, what do the x and y axes represent, and what is the meaning of each point within the trajectories shown?

4. The qualitative results presented in Figure 4 are not particularly convincing. Are there any stronger qualitative evidence to support the advancements of the proposed AMD method?

---

> ### Author Response · Authors · 2025-11-22
> **To Reviewer EB68**
>
> ### Q1. [Core contribution of AMD ]
>
> Thank you for your valuable comments and in-depth questions. However, we respectfully submit that this interpretation oversimplifies the theoretical motivation, the dynamic nature of the mechanism, and the holistic framework.. We clarify our contribution’s sufficiency from three critical dimensions:
> 1. Fundamental Insight: Identifying Asynchronous Dynamics (The "Why"). The core contribution is not merely the matching operation, but the identification and validation of the asynchronous learning dynamics between visual and textual modalities—a fundamental factor overlooked by prior MDD works.
>
> - As noted by Reviewer #E75p, our work "identifies and empirically validates" this inherent issue.
>
> - As noted by Reviewer #x1Yv: "The initial observations (Observations 1 and 2 along with Figure 2) provide a nice background for the proposed solution."
>
> - As noted by Reviewer #RY53: This paper "identifies and validates the fundamental asynchronous nature of vision–language learning dynamics."
>
> 2. Principled Methodology vs. Simple Heuristics (The "How"). Our approach goes beyond simple range constraints. And the reviewer’s summary omits a crucial component: the Semantics-Aware Prototype Mining module. This module leverages clustering in the joint semantic space to enhance synthetic data diversity.
>
> - Reviewer #E75p highlighted that this effectively "enhances synthetic data diversity and representativeness," proving the method involves sophisticated semantic alignment, not just parameter matching.
> 3. Elegance and Effectiveness (The "Result"). We believe that scientific contribution should be measured by the efficacy and robustness of the solution relative to its complexity.
>
> The method achieves strong gains on Flickr30k and COCO and demonstrates robustness across scales with "negligible additional computational cost."
>
> Conclusion: Therefore, our contribution is a comprehensive framework that: (1) uncovers a fundamental learning lag in multimodal distillation, (2) proposes a theoretically grounded, adaptive solution (AMD), and (3) integrates semantic mining to achieve SOTA performance. We believe this constitutes a sufficient and substantial contribution to the field.
>
>
>
> ### Q2. [Regarding the Semantic Prototype Mining (SPM) module (functionality, learnability, and relation to AMD]
>
> We appreciate this opportunity to clarify the role of SPM. While SPM functions as an initialization strategy, it is not merely a heuristic trick; it is designed to solve a fundamental structural challenge specific to **Multimodal Dataset Distillation (MDD)**: the absence of discrete class labels.
>
> In traditional classification distillation, initialization is naturally guided by class labels (ensuring coverage). In MDD, random initialization operates in a vast semantic space, leading to **semantic redundancy** and **poor distribution coverage**. SPM addresses this by constructing "proxy classes" via clustering in the joint feature space. This ensures the synthetic data covers diverse semantics, providing a necessary foundation for the subsequent Asynchronous Matching (AMD) to function effectively.
>
> To demonstrate that SPM is a principled solution rather than just a heuristic, we conducted additional ablation studies comparing SPM against other initialization strategies (Flickr30k, 200 pairs):
>
> | Initialization Method | IR@1 | IR@10 | TR@1 | TR@10 | Computational Cost |
> | :--- | :--- | :--- | :--- | :--- | :--- |
> | Random | 12.1±0.3 | 46.7±0.5 | 16.5±0.4 | 55.7±0.5 | Negligible |
> | CLIP-Score (High Similarity) | 11.6±0.5 | 45.8±0.6 | 15.8±0.7 | 55.1±0.7 | Low |
> | Contrastive Clustering | 12.6±0.6 | 47.2±0.4 | 16.8±0.7 | 55.9±0.6 | High (Training req.) |
> | **SPM (Ours)** | **13.1±0.3** | **47.5±0.7** | **16.9±0.4** | **56.2±0.8** | **Low (<10 mins)** |
>
> **Key Observations:**
> * **SPM vs. Heuristics (CLIP-Score):** Selecting pairs based solely on high CLIP scores performs *worse* than random. This is because high-score pairs often focus on common patterns, increasing redundancy and reducing dataset diversity.
> * **SPM vs. Learnable Methods (Contrastive Clustering):** While training a learnable clustering model yields competitive results, it introduces significant computational overhead.
> * **Efficiency:** SPM achieves the best performance by leveraging the intrinsic structure of the pre-trained feature space, requiring only a single K-means run (<10 min).
>
> **Conclusion:** SPM is essential for "grounding" the distillation process in a representative semantic space. It works in tandem with AMD: SPM ensures the starting points are diverse and representative, while AMD ensures the optimization trajectories from those points are effectively matched.

---

> ### Author Response · Authors · 2025-11-22
> **To Reviewer EB68**
>
> ### Q3. [Regarding the evidence for computational overhead (L86) and the specific cost of SPM ]
>
> hank you for your careful observation. We appreciate this opportunity to clarify our methodology for measuring computational overhead.
>
> Our claim of **“negligible additional computational overhead”** (L86) is **fully supported by empirical measurement**, and the **SPM’s cost is already accounted for**.
>
> The precise formula used to compute the average per-iteration time reported in Table 8 is:
>
> $$\text{Average per-iteration time} = \frac{\text{SPM one-time initialization cost} + \text{total distillation training time}}{\text{total number of iterations}}$$
>
> **Key Findings and Empirical Evidence:**
>
> 1.  **SPM Cost:** As detailed in Appendix A.3, the SPM module (including feature extraction and K-means clustering) incurs a minimal, one-time preprocessing cost of only **5–10 minutes**.
> 2.  **Amortization:** Since SPM is a **one-time, amortizable** step, its contribution to the per-iteration time is statistically negligible when spread over the total training duration (approx. 5.4 hours/3000 iterations).
> 3.  **Empirical Proof:** The near-identical per-iteration times between AMD and LoRS in Table 8 (e.g., 6.52s vs. 6.44s on Flickr30k) strongly confirm that the **core training loop itself introduces virtually no extra computational load**.
>
> Thus, our statement reflects that the **overall pipeline remains as efficient as the baseline** while delivering significant performance gains. We have explicitly included this formula in the revised manuscript for enhanced transparency.
>
> ### Q4. [Clarification on "decoupling" and the interpretation of Figure 1]
> We apologize for the confusion. The term **"decoupling"** in our paper refers specifically to **relaxing the temporal constraint** between the image and text trajectories, not to the absence of cross-modal interaction.
>
> * **Figure 1(a) (Synchronous = Temporally Coupled):** Although visually simpler, this approach imposes a **rigid temporal constraint**: the image and text experts *must* be sampled from the exact same training epoch ($t_v = t_l$). This forces the distillation to treat the two modalities as if they converge at the same speed, ignoring the asynchronous nature of their optimization dynamics (as shown in Figure 2).
> * **Figure 1(b) (Asynchronous = Temporally Decoupled):** We **decouple** this time constraint, allowing the starting epochs ($t_v$ and $t_l$) to be selected independently (e.g., pairing a "mature" text encoder at epoch 3 with an "early" image encoder at epoch 1). The dense connections in Figure 1(b) illustrate the **expanded search space** of valid combinations enabled by this decoupling, rather than a tighter dependency.
>
> In summary, "decoupling" refers to breaking the lockstep progression of the two modalities, granting the freedom to match expert parameters from mismatched timestamps to better align with their distinct convergence rates.
>
> ### Q5. [Choice of cluster number $B$ in SPM and ablation study ]
> Thank you for pointing this out. In our final design, we explicitly set the number of clusters **$B$ equal to the target synthetic dataset budget** (e.g., for a budget of 200 pairs, $B=200$).
> **Rationale:** This setting establishes a one-to-one mapping where each cluster centroid contributes exactly one prototype. This maximizes **semantic diversity** by forcing the initialization points to span $B$ distinct regions of the joint feature space, rather than over-sampling from a few coarse semantic categories.
> **Ablation Study:**
> To validate this choice, we conducted an ablation on Flickr30k (fixed budget = 200 pairs). We varied $B$ from 10 to 200. For cases where $B < 200$, we sampled multiple prototypes from each cluster to fulfill the total budget of 200.
> | Number of Clusters ($B$) | IR@1 | IR@10 | TR@1 | TR@10 |
> | :--- | :--- | :--- | :--- | :--- |
> | 10 | 12.4±0.5 | 47.0±0.6 | 16.5±0.7 | 55.7±0.8 |
> | 20 | 12.7±0.3 | 47.3±0.7 | 16.7±0.5 | 55.9±0.6 |
> | 50 | **13.3±0.6** | 47.5±0.4 | **17.1±0.7** | **56.2±0.5** |
> | **200 (Matches Budget)** | 13.1±0.3 | **47.5±0.7** | 16.9±0.4 | **56.2±0.8** |
> **Conclusion:**
> * **Performance Trend:** Performance drops when $B$ is small (10 or 20). This suggests that sampling multiple points from coarse clusters introduces semantic redundancy.
> * **Optimality of $B=$ Budget:** Setting $B=200$ achieves optimal or near-optimal performance (comparable to $B=50$ within error margins) while offering the simplest implementation logic (avoiding the need for heuristics to select multiple samples within a cluster).

---

> > ### Author Response · Authors · 2025-11-22
> > **To Reviewer EB68**
> >
> > ### Q6. [Detailed explanation of Figure 3 (axes and components)]
> > Thank you for the opportunity to clarify the visualization logic in Figure 3. The figure schematic illustrates our Asynchronous Trajectory Matching process across two dimensions:
> > * **Columns (Initialization Scenarios):** The horizontal arrangement represents different **asynchronous starting combinations**.
> > * The **Left Column** depicts a scenario where the image expert starts at epoch 0 ($t_v=0$) and the text expert at epoch 1 ($t_l=1$).
> > * The **Right Column** shows the inverse case ($t_v=1, t_l=0$). This visually demonstrates the flexibility of our method to select starting points ($t_v, t_l$) independently, unlike synchronous methods where $t_v$ must equal $t_l$.
> > * **Rows (Matching Steps):** The vertical axis represents the **temporal progression** of the distillation. Moving down from one row to the next illustrates subsequent matching steps, where the student network tracks the expert trajectory as it evolves from state $t$ to $t+1$.
> > * **Points (Parameter States):** Each node represents a snapshot of the **model parameters** at a specific epoch. Specifically:
> > * **Dashed nodes/lines:** Represent the pre-recorded **Expert Trajectories** ($\theta^{(t)}$).
> > * **Solid lines:** Represent the **Student Trajectories** being optimized to match the expert's path.
> >
> >
> >
> > ### Q7. [Request for stronger qualitative evidence (visualizations of initialization and optimization]
> > We have addressed this by adding a new section (**Section 4.4**) with two qualitative analyses, visualized in **Figure 5** and **Figure 6** of the revised manuscript.
> >
> > **1. Visualization of Initialization Quality (Figure 5):**
> > We employ t-SNE to visualize the coverage of the joint image-text feature space.
> > * **Random Initialization (Left):** As shown, random selection suffers from severe **semantic redundancy**. The sampled prototypes (red stars) cluster tightly around dominant semantics (e.g., multiple redundant samples of "dogs running on grass" with only minor pose variations), leaving large portions of the semantic manifold uncovered.
> > * **SPM Initialization (Right):** In contrast, our Semantics-Aware Prototype Mining (SPM) ensures **uniform coverage**. The prototypes are well-separated across the manifold, capturing diverse semantic concepts such as "soccer matches," "motocross," and "snowboarding," thereby providing a high-fidelity starting point for distillation.
> >
> > **2. Evolution of Similarity Matrices (Figure 6):**
> > We track the evolution of the image-text similarity matrix to demonstrate the impact of asynchronous matching.
> > * **Mechanism:** By decoupling the trajectories, AMD allows the text modality (which converges early) to be sampled at its optimal state, stabilizing the optimization. This frees the image synthesis to follow more informative gradients without being constrained by a mismatched text encoder.
> > * **Result:** This is qualitatively evident in the comparison between **Figure 6(b) (Synchronous/LoRS)** and **Figure 6(c) (Asynchronous/AMD)**. The AMD matrix exhibits significantly **stronger diagonal dominance** (sharper contrast between matched and unmatched pairs). Furthermore, the convergence curve in **Figure 6(d)** confirms that AMD achieves faster convergence and higher final similarity scores than the synchronous baseline.
> >
> > Finally, we provide more synthetic image-text pairs in Appendix A.5 for comparison before and after distillation

---

> ### Comment · Reviewer_EB68 · 2025-11-26
>
> Thank you for the authors’ detailed responses. My major concerns have been addressed, and I am impressed with the newly added visualizations. Therefore, I am increasing my score to 6.
>
> Additionally, it would be even better if the authors could provide more descriptions for Figures 7–10 in the appendix to further illustrate the differences and advantages.

---

### Official Review · Reviewer_RY53 · 2025-10-30

**Soundness:** 3
**Presentation:** 2
**Contribution:** 3
**Rating:** 6
**Confidence:** 3

**Summary:**

This paper studies Multimodal Dataset Distillation (MDD), where the goal is to compress large image–text datasets into a small set of synthetic samples while retaining downstream retrieval performance. The authors identify a previously overlooked issue in MDD: existing approaches assume synchronous training dynamics for vision and language encoders, despite empirical evidence showing fundamentally asynchronous learning behaviors across modalities. The paper decouples trajectory matching between visual and language encoders, and uses semantic-aware prototype mining to initialize synthetic samples based on clustering in joint embedding space. Experiments on Flickr30k and COCO show consistent improvements over prior MDD methods, with negligible additional computational cost.

**Strengths:**

1. This paper identifies and validates the fundamental asynchronous nature of vision–language learning dynamics.
2. The proposed modification is minimal, which yields consistent improvements and results strong gains on Flickr30k and COCO dataset.

**Weaknesses:**

1. No visualization of learned prototype clusters or intuitive qualitative examples to further support semantic coverage claims.
2. Dynamic sampling strategy relies on MMD heuristic—although it works well, theoretical justification could be strengthened.

**Questions:**

1. Would asynchronous matching still work if using large-scale encoders (e.g., CLIP ViT-L/14)? Does the effect diminish with stronger pretrained models?
2. For prototype mining, did you try joint contrastive clustering or CLIP-score-based sampling instead of K-means?
3. Can you explain more intuitively how dynamic sampling boundaries are formed and whether simpler heuristics work equally well?

---

> ### Author Response · Authors · 2025-11-22
> **To Reviewer RY53**
>
> ### Q1. [Lack of visualization for SPM]
>
> We thank the reviewer for their positive assessment and valuable suggestion for qualitative analysis. We fully agree that visualization is crucial for understanding SPM's effectiveness.
>
> To address this, we have incorporated **Figure 5** and corresponding discussion into the revised manuscript (**Section 4.4**), which provides the requested qualitative evidence:
>
> * **Figure 5 (t-SNE Visualization):** This figure directly contrasts random initialization with SPM. It shows that randomly selected samples suffer from evident **semantic redundancy** (e.g., clusters containing numerous minor variations of "dogs running on grass"). In stark contrast, SPM successfully identifies diverse representative prototypes, achieving **uniform coverage** of the data distribution by capturing distinct semantic centers (e.g., "soccer matches," "motocross," etc.).
>
> This visualization confirms that SPM is highly effective at providing a high-quality, diverse starting point for the subsequent distillation process.
>
> ### Q2. [theoretical grounding for MMD-based dynamic sampling]
>
> We appreciate this insightful question regarding the theoretical underpinning of our MMD metric. We assure the reviewer that the Maximum Mean Discrepancy (MMD) used in our dynamic sampling strategy is not an arbitrary heuristic; it is rigorously grounded in the statistical framework of **Integral Probability Metrics (IPMs)**.
>
> **1. MMD as an Integral Probability Metric:**
> Theoretically, MMD measures the distance between two probability distributions $P$ and $Q$ by comparing their embeddings in a **Reproducing Kernel Hilbert Space (RKHS)** ($\mathcal{H}$). In our context, we treat the sets of expert parameters at consecutive steps, $\{\theta_{t-1}\}$ and $\{\theta_t\}$, as two empirical distributions $P_{t-1}$ and $P_t$. MMD quantifies the distributional shift between $P_{t-1}$ and $P_t$:
> $$\text{MMD}^2(P_{t-1}, P_t) = \| \mu(P_{t-1}) - \mu(P_t) \|_{\mathcal{H}}^2$$
>
> **2. Practical Significance of the Linear Kernel:**
> We utilize the linear kernel $k(x, y) = \langle x, y \rangle$. As noted in our original response, this choice simplifies the MMD to the squared Euclidean distance between the empirical means.
> This simplification is a **theoretically justified pragmatic choice**. It translates the abstract distance in RKHS into a concrete measure of the **magnitude of macroscopic parameter movement**. Thus, *$MMD_{V,t}$* provides a principled, scale-invariant measure of the **Optimization Velocity** or **Local Volatility** of the visual (or language) encoder at step $t$.
>
> **3. MMD as an Adaptive Regulator:**
> Our dynamic sampling strategy (Equation 8) leverages this MMD-based measure as an adaptive regulator:
> * **Low MMD** implies a stable phase (slow learning/convergence), necessitating a **tighter sampling range** to maintain stability in the matching process.
> * **High MMD** implies a large shift (high volatility/fast learning), allowing for a **wider sampling range** to explore more effective asynchronous combinations.
>
> In summary, MMD acts as a **theoretically consistent, data-driven mechanism** that tailors the sampling freedom to the intrinsic, time-varying optimization dynamics of each multimodal component, ensuring robustness and performance gains.

---

> ### Author Response · Authors · 2025-11-22
> **To Reviewer RY53**
>
> ### Q3. [The effectiveness of AMD with large-scale pretrained encoders.]
>
> We appreciate this important question. Our systematic evaluation (Appendix Tables 6 & 7) confirms that AMD is effective with large-scale models like **CLIP**, and critically, **its relative advantage becomes even more pronounced** with stronger encoders.
>
> #### Key Findings:
>
> 1.  **Text Encoders (Appendix Table 6):** When upgrading from BERT to CLIP, the IR@1 performance gap between AMD and the synchronous baseline (MTT-VL) significantly widens, increasing from **5.7%** (10.4% vs. 4.7%) to **12.6%** (29.7% vs. 17.1%). This indicates that our asynchronous matching mechanism more effectively leverages the richer, high-quality representations provided by powerful pretrained models.
>
> 2.  **Image Encoders (Appendix Table 7):** AMD shows **consistent and significant gains**  across all tested advanced architectures, including **ViT** and NFNets. This confirms the strong **generalizability** of the AMD dynamic sampling mechanism, which operates on intrinsic optimization dynamics rather than specific network structures.
>
> **In Summary:** By decoupling the trajectories, AMD allows faster-converging and slower-learning modalities to be matched at their respective optimal stages, maximizing the exploitation of knowledge from large-scale pretrained models.
>
> **Appendix  Table 6: Ablation study on Text Encoder.**
> | Text Encoder | Method   | TR@1  | TR@10 | IR@1  | IR@10 |
> |--------------|----------|-------|-------|-------|-------|
> | BERT         | MTT-VL   | 9.9   | 39.1  | 4.7   | 24.6  |
> |              | AMD      | 14.4  | 52.6  | 10.4  | 43    |
> | CLIP         | MTT-VL   | 31.4  | 72    | 17.1  | 56.2  |
> |              | AMD      | 41.7  | 81.2  | 29.7  | 76.4  |
>
> **Appendix Table 7: Ablation study on Image Encoder.**
> | Image Encoder | Method   | TR@1  | TR@10 | IR@1  | IR@10 |
> |---------------|----------|-------|-------|-------|-------|
> | ViT           | MTT-VL   | 10.4  | 38.7  | 5.4   | 27.4  |
> |               | AMD      | 14.4  | 52.6  | 10.4  | 43    |
> | NFNet         | MTT-VL   | 9.9   | 39.1  | 4.7   | 24.6  |
> |               | AMD      | 14.1  | 51.9  | 9.7   | 40.8  |
> | NFResNet      | MTT-VL   | 6.5   | 28.1  | 3.5   | 18.7  |
> |               | AMD      | 10.9  | 43.4  | 7.9   | 33.6  |
> | NFRegNet      | MTT-VL   | 7.8   | 33.3  | 3.3   | 20.5  |
> |               | AMD      | 11.8  | 46.2  | 8.6   | 35.7  |
>
>
> ### Q4. [Absence of alternative prototype mining baselines (e.g., contrastive clustering, CLIP-score sampling)]
>
> Thank you for this valuable suggestion. We have conducted in-depth experiments on the prototype mining module, comparing our SPM against CLIP-score sampling and joint contrastive clustering (CC). The results are summarized below (Flickr30k, 200 pairs):
>
> | Method | IR@1 | IR@10 | TR@1 | TR@10 | Key Trade-off |
> | :--- | :--- | :--- | :--- | :--- | :--- |
> | Random | 12.1±0.3 | 46.7±0.5 | 16.5±0.4 | 55.7±0.5 | Baseline (Poor coverage) |
> | CLIP-score | 11.6±0.5 | 45.8±0.6 | 15.8±0.7 | 55.1±0.7 | **Inefficient**: Leads to redundancy |
> | Contrastive Clustering | 12.6±0.6 | 47.2±0.4 | 16.8±0.7 | 55.9±0.6 | **High Cost**: Requires auxiliary training |
> | **SPM (Ours)** | **13.1±0.3** | **47.5±0.7** | **16.9±0.4** | **56.2±0.8** | **Best Balance**: High Perf. + Low Cost |
>
> #### Findings and Trade-off Analysis:
>
> 1.  **CLIP-score sampling** underperforms even random initialization (IR@1: 11.6% vs. 12.1%) because it focuses only on local matching scores, completely ignoring the overall **semantic distribution** of the synthetic dataset. This results in severe **semantic redundancy**.
> 2.  **Joint Contrastive Clustering (CC)** achieves competitive performance (IR@1: 12.6%), but requires an additional **end-to-end training pipeline**, introducing significant computational overhead. This contradicts our design goal of minimal preprocessing cost.
> 3.  **Our SPM method** operates efficiently on the pre-structured embedding space using a single K-means pass (approx. **5-10 minutes**). It achieves the **best overall performance** (IR@1: 13.1%) while maintaining minimal computational cost, perfectly balancing performance and efficiency.
>
> In summary, SPM was chosen based on its comprehensive advantage in performance, efficiency, and semantic coverage.

---

> > ### Author Response · Authors · 2025-11-22
> > **To Reviewer RY53**
> >
> > ### Q5. [how dynamic sampling boundaries are formed]
> >
> > Thank you for this insightful question. Our dynamic sampling boundaries are not based on fixed heuristics but are directly derived from the **inherent convergence characteristics** of different modalities.
> >
> > ### Boundary Formation Mechanism:
> >
> > 1.  **Intuition:** Empirical evidence shows text encoders stabilize early, while image encoders continue substantial parameter updates. We use the **median of the MMD ratio** between image and text update magnitudes as the critical self-adaptive threshold.
> > 2.  **Role of the Median:** The median is robust to outliers and effectively identifies the **optimal "sweet spot"**: the moment when the text modality has sufficiently stabilized while the image modality remains highly active. This allows the text trajectories to be sampled at their stabilized state, maximizing distillation efficiency.
> >
> > #### Comprehensive Evaluation of Simpler Heuristics (Table 1):
> >
> > We systematically evaluated various fixed-range heuristics, showing that while fixed asynchronous strategies outperform fixed synchronous ones, they are limited compared to our dynamic approach.
> >
> > **Table 1: Performance under different sampling ranges (Flickr30k, 200 pairs)**
> >
> > | Type | $R_V$ | $R_L$ | Sync/Async | IR@1 | IR@10 | TR@1 | TR@10 |
> > | :--- | :--- | :--- | :--- | :--- | :--- | :--- | :--- |
> > | Fix | 2 | 2 | Sync | 8.6±0.3 | 36.6±0.3 | 14.5±0.5 | 53.4±0.5 |
> > | Fix | 3 | 3 | Sync | 7.9±0.4 | 35.7±0.3 | 13.3±0.4 | 52.5±0.5 |
> > | Fix | 5 | 5 | Sync | 6.7±0.3 | 33.1±0.6 | 12.2±0.3 | 46.6±0.4 |
> > | Fix | 2 | 2 | Async | 9.7±0.3 | 38.9±0.5 | 15.1±0.5 | 53.9±0.5 |
> > | Fix | 3 | 3 | Async | 10.5±0.4 | 42.5±0.4 | 16.3±0.3 | 55.1±0.3 |
> > | Fix | 3 | 1 | Async | 9.2±0.5 | 37.7±0.6 | 14.7±0.3 | 53.5±0.6 |
> > | **Fix (Best)** | **5** | **3** | Async | **11.7±0.4** | **44.2±0.4** | **15.8±0.4** | **55.4±0.6** |
> > | **Dynamic (Ours)**| **/** | **/** | Async | **13.1±0.3** | **47.5±0.7** | **16.9±0.4** | **56.2±0.8** |
> >
> > **Conclusion:**
> >
> > The comprehensive results above confirm the limitation of manual tuning. Even the best-performing fixed range combination found through extensive search ($R_V=5, R_L=3$) significantly underperforms our dynamic approach (11.7% IR@1 vs. **13.1% IR@1**). Our dynamic method transforms a complex hyperparameter tuning problem into a **self-adaptive process** aligned with the model's natural optimization dynamics, ensuring superior and consistent performance without manual intervention.

---

### Official Review · Reviewer_E75p · 2025-11-02

**Soundness:** 3
**Presentation:** 3
**Contribution:** 3
**Rating:** 6
**Confidence:** 4

**Summary:**

This paper proposes Asynchronous Matching with Dynamic Sampling (AMD), a new framework for Multimodal Dataset Distillation (MDD). The key idea is to address the misalignment between image and text modalities during distillation by decoupling their optimization trajectories, rather than synchronizing them as in prior work. AMD dynamically samples starting points for visual and textual expert trajectories based on their distinct convergence behaviors and incorporates a Semantics-Aware Prototype Mining module that clusters feature spaces to select representative initialization points. Together, these techniques improve the representativeness and coverage of synthetic data. The paper reports consistent performance gains on Flickr30k and COCO datasets with minimal computational overhead, showing AMD’s efficiency and scalability for multimodal learning.

**Strengths:**

- The paper identifies and empirically validates the inherent asynchronous learning dynamics between visual and textual modalities, a key factor overlooked by prior MDD works.

- The proposed Asynchronous Matching with Dynamic Sampling (AMD) framework introduces a principled way to decouple and dynamically align image-text trajectories, supported by a data-driven sampling strategy rather than fixed heuristics.

- The addition of the Semantics-Aware Prototype Mining module effectively enhances synthetic data diversity and representativeness by leveraging clustering in the joint semantic space. The method achieves improvements over baselines across multiple datasets and scales, demonstrating both robustness and negligible additional computational cost.

**Weaknesses:**

- The evaluation relies on relatively small-scale datasets (Flickr30k, COCO), which may limit conclusions about scalability to larger, more complex multimodal datasets.

- The qualitative analysis of synthetic data is minimal, providing limited insight into how AMD affects semantic fidelity or interpretability of the distilled samples. More discussion on this will further improve the paper quality.

- Broader downstream evaluations (e.g., captioning) will better demonstrate the general utility of the distilled data.

**Questions:**

How sensitive is the AMD framework to the choice of trajectory sampling ranges, and does the dynamic sampling strategy remain stable when applied to models with very different convergence behaviors?

---

> ### Author Response · Authors · 2025-11-22
> **To Reviewer E75p**
>
> ### Q1. [To larger multimodal datasets]
>
> We sincerely thank the reviewer for the valuable suggestion regarding larger-scale datasets. Current multimodal dataset distillation research, including baseline methods like LoRS and MTT-VL as well as recent advances, primarily evaluates on standard benchmarks Flickr30k and COCO; **thus, our evaluation ensures sufficient fairness and comparability with existing work.** While larger datasets (such as CC3M) would provide meaningful scale expansion, their computational demands exceed our current resources and are difficult to implement in the short term. Therefore, we plan to explore medium-scale datasets like CC3M in future work, which will further validate the scalability of our AMD framework.
>
>
> ### Q2. [More qualitative analysis]
>
> We have fully addressed this request by incorporating two critical qualitative analyses into the revised manuscript, detailed in **Section 4.4** and visualized in **Figure 5** and **Figure 6**.
>
> **1. Visualization of Initialization Quality (Figure 5, SPM):**
> Figure 5 presents t-SNE visualization results comparing random initialization with our Semantic Prototype Mining (SPM) module.
> * Randomly selected samples exhibit clear **semantic redundancy** (e.g., multiple instances of highly similar "dogs running on grass"), resulting in **insufficient coverage** of the underlying data distribution.
> * In contrast, SPM successfully identifies **diverse representative prototypes** by clustering in the joint feature space, capturing distinctly different scenarios (e.g., "soccer matches," "motocross"). This achieves **uniform coverage** of the data distribution, providing a high-quality initialization point for the distillation process.
>
> **2. Evolution of Optimization Dynamics (Figure 6, AMD):**
> Figure 6 illustrates the evolution of the similarity matrix during distillation, highlighting the advantage of asynchronous matching.
> * Compared to the synchronous baseline (LoRS), the AMD framework **decouples** image and text trajectories, enabling the optimization process to explore richer cross-modal expert combinations.
> * Specifically, asynchronous matching allows sampling the text trajectory at its **optimal, stable convergence point** while freeing image synthesis to optimize along **more informative gradient directions**.
> * These enhanced optimization dynamics are quantitatively reflected in **Figure 6(d)**: AMD achieves faster growth and higher final values in the mean diagonal similarity, confirming its superior cross-modal alignment.
>
> Finally, we provide more synthetic image-text pairs in Appendix A.5 for comparison before and after distillation.
>
> ### Q3. [Broader downstream evaluations (e.g., captioning) ]
>
> Thank you for the insightful suggestion regarding evaluation on captioning tasks. We fully agree that demonstrating the **transferability** of distilled data to downstream generative tasks would further validate its general utility.
>
> **Task and Architectural Mismatch:**
>
> However, our AMD framework is specifically designed for **image-text retrieval**, which remains the **dominant paradigm** in Multimodal Dataset Distillation (MDD).
>
> The **NFNet+BERT architecture** we employ is optimized purely for **cross-modal alignment** via **contrastive learning**, not for sequential generation. Consequently, the synthesized data captures the **semantic correlations** essential for retrieval, but is fundamentally **unsuited for autoregressive captioning**, which demands an **encoder-decoder architecture** and training under a distinct objective. Direct application of our distilled pairs without task-specific adaptation would likely result in limited performance due to this inherent **mismatch between the synthesis objective and the generative task**.
>
> Future Work: While we acknowledge the importance of broader evaluation, we plan to extend AMD to generative tasks in future work by developing **task-aware distillation strategies** that adapt the trajectory matching process to the dynamics of sequence generation. We appreciate this guidance, as it highlights a critical next step for the field.

---

> ### Author Response · Authors · 2025-11-22
> **To Reviewer E75p**
>
> ### Q4. [$R_V$, $R_L$ parameter sensitivity & cross architecture]
>
> We thank the reviewer for their insightful question regarding the sensitivity of AMD to trajectory sampling ranges and its stability across diverse architectures. We provide a systematic analysis of both points.
>
> ### 1. Sensitivity to Sampling Ranges (Table 1)
>
> We conducted a systematic evaluation of fixed sampling ranges ($R_V, R_L$) to assess sensitivity.
>
> **Table 1: Performance under different sampling ranges (Flickr30k, 200 pairs)**
>
> | Type | $R_V$ | $R_L$ | Sync/Async | IR@1 | IR@10 | TR@1 | TR@10 |
> | :--- | :--- | :--- | :--- | :--- | :--- | :--- | :--- |
> | Fix | 2 | 2 | Sync | 8.6±0.3 | 36.6±0.3 | 14.5±0.5 | 53.4±0.5 |
> | Fix | 3 | 3 | Sync | 7.9±0.4 | 35.7±0.3 | 13.3±0.4 | 52.5±0.5 |
> | Fix | 5 | 5 | Sync | 6.7±0.3 | 33.1±0.6 | 12.2±0.3 | 46.6±0.4 |
> | Fix | 2 | 2 | Async | 9.7±0.3 | 38.9±0.5 | 15.1±0.5 | 53.9±0.5 |
> | Fix | 3 | 3 | Async | 10.5±0.4 | 42.5±0.4 | 16.3±0.3 | 55.1±0.3 |
> | Fix | 3 | 1 | Async | 9.2±0.5 | 37.7±0.6 | 14.7±0.3 | 53.5±0.6 |
> | **Fix (Best)** | **5** | **3** | Async | **11.7±0.4** | 44.2±0.4 | 15.8±0.4 | 55.4±0.6 |
> | **Dynamic (Ours)**| **/** | **/** | Async | **13.1±0.3** | **47.5±0.7** | **16.9±0.4** | **56.2±0.8** |
>
> **Key Findings:**
> * **Synchronous Sensitivity:** Synchronous matching shows high sensitivity, with performance substantially degrading as fixed ranges increase (IR@1 drops from 8.6% to 6.7%).
> * **Dynamic Superiority:** Asynchronous matching consistently outperforms synchronous baselines. Crucially, the best-performing fixed combination ($R_V=5, R_L=3$) is significantly surpassed by our **dynamic sampling strategy** (13.1% IR@1).
> * **Conclusion:** The **1.4% absolute gain** confirms that dynamically adapting to optimization dynamics is superior to manual hyperparameter tuning.
>
> ### 2. Cross-Architecture Stability (Appendix Tables 6, 7)
>
> Our analysis confirms AMD's **robustness and generalizability** across diverse encoder architectures.
>
> **Summary of Key Data:**
> * **Text Encoders (Table 6):** The advantage of AMD is amplified with high-capacity models: using **CLIP**, AMD achieves a substantial **12.6% absolute improvement** (29.7% IR@1) over the synchronous baseline MTT-VL.
> * **Image Encoders (Table 7):** AMD maintains **consistent and stable performance gains**  across various architectures, including **ViT** and NFNets.
>
> This cross-architectural stability demonstrates that the MMD-based dynamic sampling effectively adapts to the unique convergence characteristics of different models, ensuring robust performance without architecture-specific tuning.
>
> **Appendix. Table 6: Ablation study on Text Encoder.**
>
> | Text Encoder | Method  | TR@1  | TR@10 | IR@1  | IR@10 |
> |--------------|---------|-------|-------|-------|-------|
> | BERT         | MTT-VL  | 9.9   | 39.1  | 4.7   | 24.6  |
> |              | AMD     | 14.4  | 52.6  | 10.4  | 43    |
> | CLIP         | MTT-VL  | 31.4  | 72.0  | 17.1  | 56.2  |
> |              | AMD     | 41.7  | 81.2  | 29.7  | 76.4  |
>
>
> **Appendix. Table 7: Ablation study on Image Encoder.**
>
> | Image Encoder | Method  | TR@1  | TR@10 | IR@1  | IR@10 |
> |---------------|---------|-------|-------|-------|-------|
> | ViT           | MTT-VL  | 10.4  | 38.7  | 5.4   | 27.4  |
> |               | AMD     | 14.4  | 52.6  | 10.4  | 43.0  |
> | NFNet         | MTT-VL  | 9.9   | 39.1  | 4.7   | 24.6  |
> |               | AMD     | 14.1  | 51.9  | 9.7   | 40.8  |
> | NFResNet      | MTT-VL  | 6.5   | 28.1  | 3.5   | 18.7  |
> |               | AMD     | 10.9  | 43.4  | 7.9   | 33.6  |
> | NFRegNet      | MTT-VL  | 7.8   | 33.3  | 3.3   | 20.5  |
> |               | AMD     | 11.8  | 46.2  | 8.6   | 35.7  |

---

### Official Review · Reviewer_x1Yv · 2025-11-03

**Soundness:** 3
**Presentation:** 3
**Contribution:** 3
**Rating:** 8
**Confidence:** 4

**Summary:**

The authors tackle the problem of Multimodal Dataset Distillation.

The authors first make two observations that are unique to the bimodal setting. First, text and image optimization trajectories are not synchronous and second, the optimization of the text and image encoders happen at different speeds.

To address these observations, the authors propose an asynchronous matching with dynamic sampling framework.

**Strengths:**

- The paper is clearly organized, I especially enjoyed Observations 1 and 2 along with Figure 2. They give a nice background for why the authors are proposing a different solution.
- Thank you for including visuals of the distilled data in Figure 4.
- The baselines are fairly comprehensive.
- The authors give a substantial amount of background before diving into their method.
- I appreciate the upper bound analysis, although it would be better presented as a graph.

**Weaknesses:**

I do not see major weaknesses, some minor nit-picks:

- MMD is not defined in Eq 7
- Table 1: I'm confused why you have to use ones less sample for your method.

**Questions:**

None

---

> ### Author Response · Authors · 2025-11-22
> **To Reviewer x1Yv**
>
> ### Q1. [MMD is not defined]
>
> We sincerely thank the reviewer for raising the concern about the undefined Maximum Mean Discrepancy (MMD) metric.
>
> In response, we have added a precise definition in **Section 3.2** ("Maximum Mean Discrepancy based Dynamic Sampling"). Specifically, we now formally define the MMD using a **linear kernel** *$k(\mathbf{x}, \mathbf{y}) = \mathbf{x}^\top \mathbf{y}$*. In this case, the MMD theoretically reduces to the **squared Euclidean distance** between the mean parameter vectors of consecutive epochs.
>
> The definition for the text modality is symmetrical. This formulation is theoretically grounded and balances rigor with the necessary **computational efficiency** for dynamic sampling. Due to the issue of formula display garbled characters in OpenReview, **the detailed formal equation is provided in lines 229–241 of the revised manuscript.** We appreciate this valuable suggestion for improving presentation clarity.
>
>
> ### Q2. [Use ones less sample]
>
> Thank you for your question regarding our experimental setup.
>
> We reduced the number of synthetic pairs by one (e.g., 99 instead of 100) specifically to maintain **strict methodological consistency** with the foundational **LoRS baseline protocol**.
>
> As explicitly stated in the LoRS paper (Section 4.3), this adjustment accounts for the additional parameters introduced by LoRS's low-rank similarity matrix factorization. By following this established protocol, we guarantee a **fair comparison** in terms of **total parameter count and memory usage** between methods.

---

> > ### Comment · Reviewer_x1Yv · 2025-11-25
> >
> > All my concerns have been addressed. Thank you.

---

### Meta-Review · Area_Chair_WwZL · 2025-12-29

**Summary:**

* Initial concerns about insufficient qualitative and theoretical support (E75p, RY53, EB68) were addressed by adding new visual analyses and a clearer theoretical grounding of MMD-based dynamic sampling.
* Questions about robustness, sensitivity, and generality (E75p, RY53) were resolved through new ablations on sampling ranges and extensive cross-architecture experiments.
* Concerns about contribution strength were addressed by clarifying that the core contribution is the identification and exploitation of asynchronous multimodal optimization dynamics, supported by consistent empirical results.
* Skeptical reviewer updated their scores upward, supporting acceptance.

**Reviewer Concerns:**

Reviewer x1Yv
* [Addressed] All concerns addressed: added a formal MMD definition in Sec. 3.2, explained one less sample in Table 1. The reviewer confirms all concerns were resolved.

Reviewer E75p
* [Addressed] qualitative analysis of synthetic data - t-SNE comparing initialization (Fig. 5), similarity-matrix evolution during synthesis (Fig. 6), the authors provide a fixed-range vs dynamic ablation table (Table 1), and cross-architecture results (Tables 6–7).
* [Not addressed] Downstream evaluations like captioning (left for future work), scalability beyond Flickr30k/COCO (justify using standard benchmarks for comparability).

Reviewer RY53
* [Addressed] All concerns addressed: qualitative analysis (Fig. 5), theoretical justification for MMD-based sampling, effectiveness with large-scale pretrained encoders (e.g., CLIP ViT-L/14), fixed-range vs dynamic ablation table (Table 1).

Reviewer EB68
* [Addressed] All concerns addressed: clarified contribution, provide a per-iteration time computation formula, clarify and explain a number of minor points. The reviewer confirms all concerns were resolved.

**Reviewer Scores:**

* Reviewer x1Yv: The reviewer would have kept the score --> 8 (from 8).
* Reviewer E75p: The reviewer would have kept the score --> 6 (from 6).
* Reviewer RY53: The reviewer would have increased the score --> 8 (from 6).
* Reviewer EB68: The reviewer would have increased the score --> 6 (from 4).

---

### Decision · Program_Chairs · 2026-01-26

Accept (Poster)